# Adipocyte-derived kynurenine promotes obesity and insulin resistance by activating the AhR/STAT3/IL-6 signaling

Teng Huang [1,6], Jia Song[1,2,6], Jia Gao[1,6], Jia Cheng[1,3,6], Hao Xie[1], Lu Zhang[1], Yu-Han Wang[1], Zhichao Gao[1], Yi Wang [1], Xiaohui Wang[1], Jinhan He[4], Shiwei Liu[5], Qilin Yu[1], Shu Zhang[1], Fei Xiong [1✉], Qing Zhou [1✉] & Cong-Yi Wang [1✉]

Aberrant amino acid metabolism is a common event in obesity. Particularly, subjects with obesity are characterized by the excessive plasma kynurenine (Kyn). However, the primary source of Kyn and its impact on metabolic syndrome are yet to be fully addressed. Herein, we show that the overexpressed indoleamine 2,3-dioxygenase 1 (IDO1) in adipocytes predominantly contributes to the excessive Kyn, indicating a central role of adipocytes in Kyn metabolism. Depletion of *Ido1* in adipocytes abrogates Kyn accumulation, protecting mice against obesity. Mechanistically, Kyn impairs lipid homeostasis in adipocytes via activating the aryl hydrocarbon receptor (AhR)/Signal transducer and activator of transcription 3 /interleukin-6 signaling. Genetic ablation of AhR in adipocytes abolishes the effect of Kyn. Moreover, supplementation of vitamin B6 ameliorated Kyn accumulation, protecting mice from obesity. Collectively, our data support that adipocytes are the primary source of increased circulating Kyn, while elimination of accumulated Kyn could be a viable strategy against obesity.

[1] Department of Respiratory and Critical Care Medicine, the Center for Biomedical Research, NHC Key Laboratory of Respiratory Diseases, Tongji Hospital, Tongji Medical College, Huazhong University of Science and Technology, Wuhan, China. [2] Reproductive Medicine Center, Tongji Hospital, Tongji Medical College, Huazhong University of Science and Technology, Wuhan, China. [3] Division of Cardiology, Department of Internal Medicine, Tongji Hospital, Tongji Medical College, Huazhong University of Science and Technology, Wuhan, China. [4] Department of Pharmacy, National Clinical Research Center for Geriatrics, West China Hospital, Sichuan University, Chengdu, China. [5] Shanxi Bethune Hospital, Shanxi Academy of Medical Sciences, Tongji Shanxi Hospital, Third Hospital of Shanxi Medical University, Taiyuan, China. [6]These authors contributed equally: Teng Huang, Jia Song, Jia Gao, Jia Cheng. ✉email: feixiong@tjh.tjmu.edu.cn; zhouqing@tjh.tjmu.edu.cn; wangcy@tjh.tjmu.edu.cn

Obesity is a leading risk factor for various metabolic diseases, including type 2 diabetes, fatty liver and cardiovascular diseases[1–3], which impairs the quality of life coupled with economic burden to families and society[4]. Moreover, the remarkable increase of obesity during the past decades indicates a significant urgency for developing more effective strategies against this devastating disorder[5].

In general, obesity is accompanied by the aberrant metabolism of amino acids[6–8]. Specifically, there is a feasible evidence that kynurenine (Kyn), a downstream catabolite of tryptophan (Trp), is upregulated in the plasma from certain subjects with obesity, which could be caused by the increased activity of indoleamine 2,3-dioxygenase 1 (IDO1)[9–11]. Overexpressed IDO1 exhausts circulating Trp and competitively inhibits the production of other Trp metabolites such as serotonin, which involves in satiety generation and appetite suppression[12,13]. Kyn is generally recognized as an immunosuppressive factor[14,15], while obesity is always accompanied by the low-grade chronic inflammation. Therefore, the increase of circulating Kyn has been considered as a compensatory effect[16,17]. However, the reality is that the increase of Kyn did not seem to ameliorate the inflammatory microenvironment in subjects with obesity[9,18], suggesting a complex role of Kyn in individuals with obesity. Moreover, the primary source of the excessive Kyn in obesity remains unexplored, leading to a lack of therapeutic strategy aiming at Kyn depletion.

Adipose tissue (AT) is considered as an energy-storage depot to maintain glucose and lipid homeostasis[19,20]. It can also function as an endocrine organ to secret multiple cytokines, such as leptin, adiponectin, and interleukin-6 (IL-6), implicated in the pathogenesis of obesity[21]. Adipocytes, the predominant cellular subpopulation of AT, play a key role in the maintenance of metabolic homeostasis[22,23]. In individuals with obesity, extensive circulating glucose is transported into adipocytes to synthesize fatty acid (FA)[24]. Excessive FAs are next esterified into triglycerides to form lipid droplets and stored in adipocytes featured by hypertrophy[24]. Those altered metabolic processes in turn prompt adipocytes to release proinflammatory cytokines, which induce or aggravate insulin resistance[25]. Unfortunately, most of the studies were focused on the metabolism of carbohydrate and lipid within the adipocytes, while alterations in amino acid metabolism in adipocytes and the related pathogenesis are less investigated.

In this study, we first identified that mature adipocytes from white adipose tissue (WAT) are critical for the metabolism of Kyn, and act as the primary headstream of increased circulating Kyn in subjects with obesity. *Ido1* deficiency in adipocytes attenuated the increase of Kyn following high-fat-diet (HFD) challenge, thereby protecting the mice from obesity. Mechanistically, the excessive Kyn-induced aryl hydrocarbon receptor (AhR) overexpression, which then transcribed Signal transducer and activator of transcription 3 (STAT3) and activated the STAT3-IL6 signaling, by which Kyn mediated a systemic effect on the development of obesity and insulin resistance. Additionally, exogenous vitamin B6 (Vit-B6) efficiently catalyzed Kyn catabolism, which remarkably reduced the amount of adipose and circulating Kyn, and rescued the HFD-fed mice from obesity. Together, our data support that over nutrition renders adipocytes to produce copious amount of Kyn, which then exacerbates insulin resistance via the AhR/STAT3/IL-6 axis, and therefore, supplementation of Vit-B6 enhanced Kyn catabolism and protected mice from HFD-induced obesity and insulin resistance. Therefore, prevention and elimination of Kyn accumulation could be a promising strategy against obesity and insulin resistance in clinical settings.

## Results

**IDO1-catalyzed Kyn exacerbates insulin resistance in subjects with obesity**. There is a suggestive evidence that females with obesity are featured by the increase of circulating Kyn[26]. To further confirm this observation, we collected 735 human plasma samples, 113 of which were derived from subjects with obesity, 268 were from subjects with overweight, and the rest 354 were from lean subjects (Supplementary Table 1). Indeed, the levels of plasma Kyn were positively correlated with BMI ($R^2 = 0.16$, $p < 0.0001$, Fig. 1a) regardless of gender, while no significant difference in terms of plasma Trp was noted between 3 groups of subjects (Supplementary Fig. 1a, b). In particular, subjects with obesity displayed significantly higher levels of circulating Kyn as compared to that of subjects with overweight ($393.85 \pm 12.20$ ng mL$^{-1}$ vs. $336.23 \pm 6.00$ ng mL$^{-1}$, $p < 0.0001$, Fig. 1b). The Kyn/Trp ratio (KTR), which reflects IDO1 activity[13], was also higher in the group affected by obesity than that in group with overweight (Fig. 1c). Similarly, mice challenged with HFD exhibited increased plasma Kyn level (Fig. 1d) and KTR (Fig. 1e and Supplementary Fig. 1c) as compared to that of mice fed with normal chow diet (NCD).

Given that Kyn has been recognized as an immunosuppressive factor[14,15], the above results prompted us to assume that the increased circulating Kyn could be a compensatory effect against chronic inflammation in individuals with obesity. We thus injected the mice subcutaneously with Kyn (20 mg kg$^{-1}$ d$^{-1}$) or PBS for 30 consecutive days starting from the eighth weeks of a 12-week period of HFD challenge. Pharmacokinetic studies were first conducted (Supplementary Fig. 1d) to demonstrate the biosafety of Kyn dosage employed above. After administrating exogenous Kyn for one month, plasma Kyn levels significantly increased (Supplementary Fig. 1e). Unexpectedly, exogenous Kyn rendered the mice gained higher body weight as compared to that of PBS-treated mice (Fig. 1f), coupled with impaired glucose tolerance (Fig. 1g and Supplementary Fig. 1f) and aggravated insulin resistance (Fig. 1h–i and Supplementary Fig. 1g). Western blot results further indicated that Kyn administration also exacerbated insulin resistance in the liver and skeletal muscle (Supplementary Fig. 1h) along with an enhanced gluconeogenesis and attenuated synthesis of hepatic glycogen (Supplementary Fig. 1i).

Since Kyn is a Trp downstream metabolite catalyzed by IDO1[27], *Ido1*$^{-/-}$ mice were next employed to dissect the exact role of Kyn in obesity. As expected, depletion of *Ido1* almost completely depleted plasma Kyn (Supplementary Fig. 2a–c). However, unlike the above exogenous Kyn studies, depletion of *Ido1* attenuated body weight gain (Fig. 1j) and improved insulin resistance induced by HFD (Fig. 1k–m and Supplementary Fig. 2d–f), while no perceptible change was noted between two groups of mice once they fed with NCD (Supplementary Fig. 2g–j). In line with these results, metabolic cage assays (Supplementary Fig. S2k, l) and positron emission tomography (PET) images (Supplementary Fig. 2m) indicated a shift of energy source towards carbohydrate utilization in *Ido1*$^{-/-}$ mice. In sharp contrast, HFD-fed WT mice exhibited abundant lipid depositions in the liver and AT along with the presence of fatty liver and adipocyte hypertrophy (Supplementary Fig. 2n, o). *Ido1*$^{-/-}$ mice displayed diminished M1 adipose tissue macrophages (Supplementary Fig. 2p) and decreased circulating inflammatory cytokines (e.g., IL-6 and IL-1β) (Supplementary Fig. 2q). Those observations are contrary to the well-known immunosuppressive function of IDO1, suggesting a more complex role of IDO1 in individuals with obesity.

A rescue study was next conducted in *Ido1*$^{-/-}$ mice by administration of exogenous Kyn as above (Supplementary

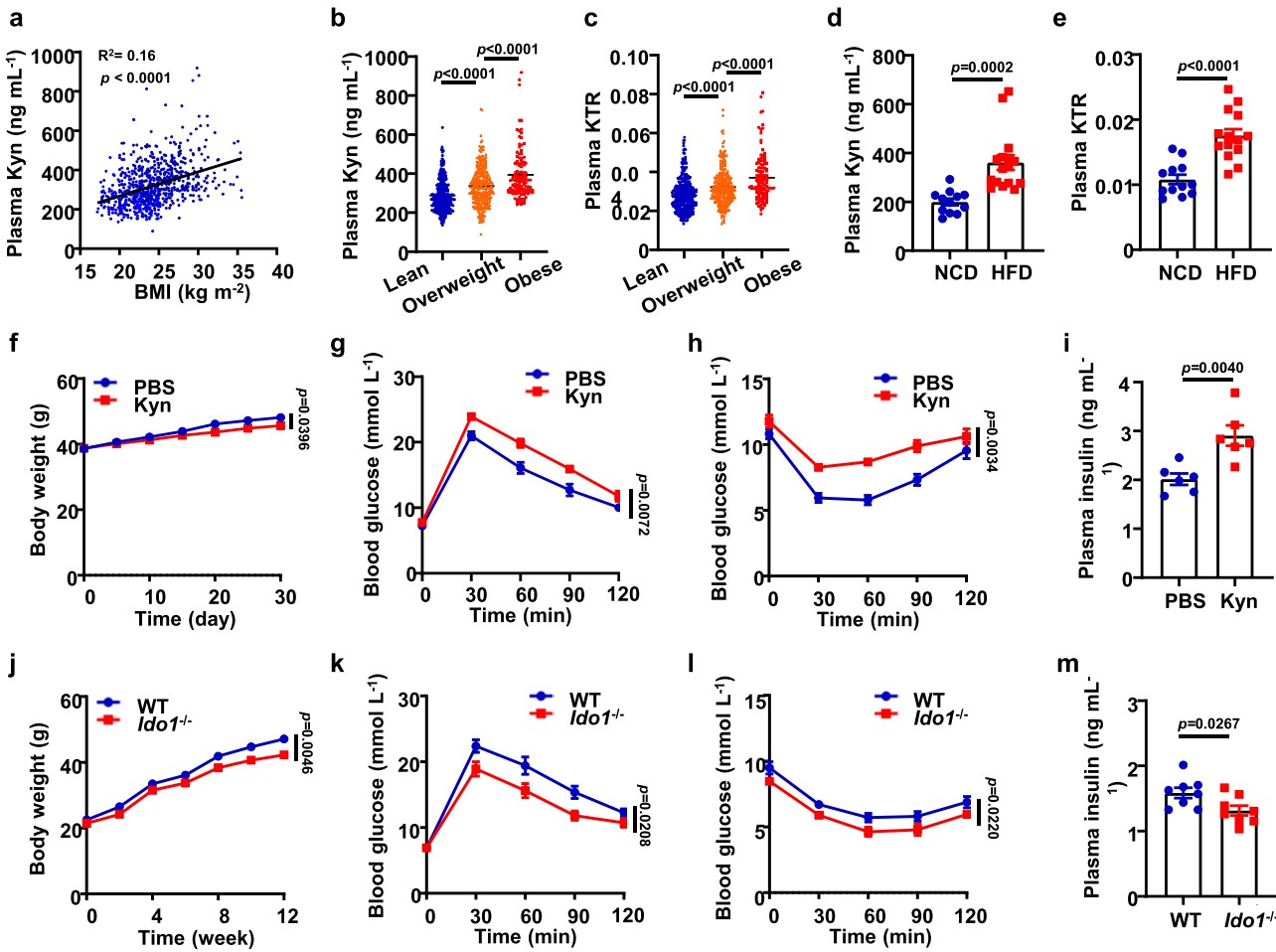

**Fig. 1 IDO1-catalyzed Kyn exacerbates insulin resistance in subjects with obesity. a** Correlation analysis between body mass index (BMI) and plasma Kyn levels in human samples ($n = 735$). Plasma Kyn levels (**b**) and KTR (**c**) in lean subjects ($18 \leq BMI < 24$, $n = 354$), subjects with overweight ($24 \leq BMI < 28$, $n = 268$) and subjects with obesity (BMI $\geq 28$, $n = 113$). **d** Plasma Kyn levels and **e** KTR in WT mice fed with NCD ($n = 12$) or HFD ($n = 16$). **f** Body weights of Kyn-treated and PBS-treated WT mice. Glucose tolerance test (GTT) (**g**) and insulin tolerance test (ITT) (**h**) in Kyn-treated and PBS-treated WT mice ($n = 6$). **i** Plasma insulin levels of Kyn-treated and PBS-treated WT mice ($n = 6$). **j** Body weights of WT and $Ido^{-/-}$ mice fed with HFD for 12 weeks ($n = 8$). GTT (**k**) and ITT (**l**) in WT and $Ido^{-/-}$ mice fed with HFD for 12 weeks ($n = 8$). **m** Plasma insulin levels of WT and $Ido^{-/-}$ mice fed with HFD for 12 weeks ($n = 8$). Data were represented as mean ± SEM. Statistical significance was assessed by two-sided Spearman's correlation (**a**), One-way ANOVA (**b, c**), two-sided Student's $t$ test (**d, e, i** and **m**) or two-way ANOVA followed with Bonferroni's multiple comparisons test (**f–h, j–l**) and significant differences were indicated with $p$ values. Source data are provided in the Source Data file.

Fig. 3a). As compared to PBS-treated $Ido1^{-/-}$ mice, Kyn significantly enhanced body weight gain (Supplementary Fig. 3b) and aggravated insulin resistance in $Ido1^{-/-}$ mice during the course of HFD induction (Supplementary Fig. 3c–e). Collectively, our data support that the high levels of circulating Kyn in subjects affected by obesity are likely caused by the enhanced IDO1 expression, and Kyn seems acting as an agonist to promote obesity and insulin resistance rather than serving as an immune-suppressor to repress chronic inflammation in the adipose tissue.

**Mature adipocytes from WAT could be the major source of plasma Kyn.** Next, we sought to characterize the cell types responsible for the altered plasma Kyn. Major metabolic organs, including white adipose tissue (WAT), liver and skeletal muscle (SK), were collected from NCD or HFD-fed mice for Kyn quantitative analysis. A remarkable increase of Kyn and KTR was only noted in the WAT of HFD-fed mice, while there was no significant difference in terms of Kyn or KTR in the liver and SK (Fig. 2a and Supplementary Fig. 4). Consistently, RT-qPCR results demonstrated that $Ido1$ was strikingly upregulated in the

WAT of HFD-fed mice, even higher than that in the spleen (Fig. 2b). To dissect the exact cellular subpopulation that contributes to augmented plasma Kyn, WAT was digested and separated by centrifugation into mature adipocytes and stromal vascular fraction (SVF). Only mature adipocytes from obese mice manifested $Ido1$ overexpression both in transcriptional and protein levels, and no perceptible difference was observed in SVF which includes abundant immune cells such as macrophages and T cells (Fig. 2c, d).

To reinforce the evidence, we further checked IDO1 expression in human AT. Similar as above, IDO1 was markedly upregulated in the WATs originated from subjects with obesity (Fig. 2e). Moreover, RT-qPCR analysis of $IDO1$ mRNA established a positive correlation between BMI and $IDO1$ transcriptional levels in adipocytes from human WATs ($n = 22$, Table 2, Supporting Information; $R^2 = 0.23$, $p < 0.05$, Fig. 2f).

Macrophages are the predominant subpopulation of immune cell in AT[28]. To further exclude the impact of IDO1 on adipose tissue macrophage (ATM), the macrophage-specific $Ido1$ knockout mice (the $LyzM$-Cre$^+$-$Ido1^{flox/flox}$ mice, hereinafter defined as $Ido1$-lKO mice) were generated by crossing the $Ido1^{flox/flox}$ mice

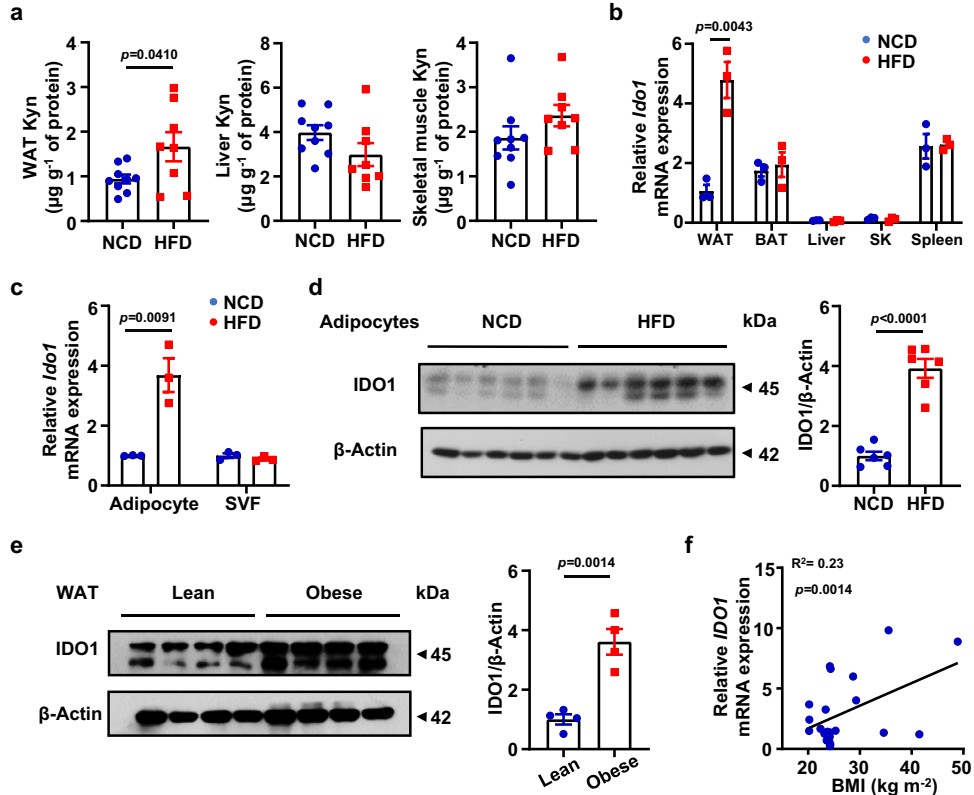

**Fig. 2 Mature adipocytes from WAT serve as the major source of plasma Kyn. a** Concentration of Kyn in the WAT, liver and skeletal muscle from NCD and HFD mice ($n = 8$–9). **b** The transcriptional level of *Ido1* in various tissues of NCD and HFD mice were assessed by RT-qPCR analysis ($n = 3$). **c** RT-qPCR analysis for *Ido1* mRNA expression in mature adipocytes and SVF originated from NCD or HFD fed mice ($n = 3$). **d** Expression level of IDO1 in mature adipocytes of NCD and HFD mice ($n = 6$). **e** IDO1 expression in visceral white adipose tissue from lean ($18 \leq BMI < 24$) and subjects with obesity ($BMI \geq 28$, $n = 4$). **f** Correlation analysis between BMI and IDO1 levels in human mature adipocytes ($n = 22$). Data were expressed as mean ± SEM. Statistical significance was assessed by two-sided Student's *t* test (**a**, **d** and **e**), two-way ANOVA (**b** and **c**) or two-sided Spearman's correlation (**f**) and significant differences were indicated with *p* values. Source data are provided in the Source Data file.

with the *LyzM*-Cre mice (Supplementary Fig. 5a), and their littermates (the *LyzM*-Cre⁻-*Ido1*^flox/flox mice, hereinafter defined as Ctrl mice) were employed as controls. Unlike the conventional *Ido*⁻/⁻ mice, the *Ido1*-lKO mice following a 12-week of HFD induction exhibited comparative bodyweight change (Supplementary Fig. 5b, c) and insulin sensitivity (Supplementary Fig. 5d, e) as their littermates, confirming that macrophages are not the critical target for IDO1 modulation of the development of obesity. Together, those data support that the mature adipocytes from WATs might be the headstream for the elevated plasma Kyn in individuals with obesity.

**Adipocyte *Ido1* deficiency renders the mice with decreased Kyn level and resistance to obesity**. The above results prompted us to assume that WAT mature adipocytes might be the headstream responsible for the excessive circulating Kyn. To address this hypothesis, an adipocyte-specific *Ido1* knockout model was generated by crossing the *Ido1*^flox/flox mice with the *Adipoq*-CreER^T2 mice (the *Adipoq*-CreER^T2-*Ido1*^flox/flox mice, hereinafter defined as *Ido1*-aKO mice), and the resulting littermates (the *Adipoq*-CreER^T2−-*Ido1*^flox/flox mice, hereinafter defined as Ctrl mice) were served as controls (Fig. 3a). In contrast to the *Ido1*-lKO mice, the *Ido1*-aKO mice were resistant to HFD-induced obesity (Fig. 3b). Metabolite assays revealed that *Ido1* deficiency in adipocytes significantly reduced the amount of Kyn and decreased KTR in the WAT, but did not affect the amount of Trp in the WAT (Fig. 3c). Consistently, the *Ido1*-aKO mice following HFD

challenge displayed significantly lower levels of plasma Kyn and KTR, but no perceptible difference in terms of plasma Trp was observed (Fig. 3d). These results provided solid evidence that mature adipocytes in the WAT are responsible for the excessive plasma Kyn in individuals with obesity.

Next, metabolic properties were examined in those HFD challenged *Ido1*-aKO mice. An enhanced carbohydrate utilization was noted in the *Ido1*-aKO mice as evidenced by the increased respiratory exchange ratio (RER), as well as an increased heat production (Fig. 3e), but they exhibited comparative food intake as their littermate controls (Supplementary Fig. 6a). More intensive uptake of ¹⁸F fluorodeoxyglucose (FDG)[29] was observed in the WAT and BAT of the *Ido1*-aKO mice, indicating an elevated carbohydrate utilization for WAT and BAT in the *Ido1*-aKO mice (Fig. 3f).

In consistent with the attenuated obesity phenotype, the HFD challenged *Ido1*-aKO mice exhibited significantly improved glucose tolerance (Fig. 3g) and insulin sensitivity (Fig. 3h–j). Additionally, steatosis was observed in the control mice as validated by the massive lipid depositions (Supplementary Fig. 6b), while the *Ido1*-aKO mice were featured by the increased lipolysis (p-HSL^Ser660) and decreased lipogenesis (p-ACC^Ser79) (Supplementary Fig. 6d) coupled with relieved adipocyte hypertrophy in the WAT (Supplementary Fig. 6c). Moreover, a remarkable increase of M2 macrophages (CD11b⁺/F4/80⁺/CD206⁺/CD11c⁻) along with a significant reduction of M1 macrophages (CD11b⁺/F4/80⁺/CD206⁻/CD11c⁺) were detected in the epididymal WAT from the *Ido1*-aKO mice (Fig. 3k), which

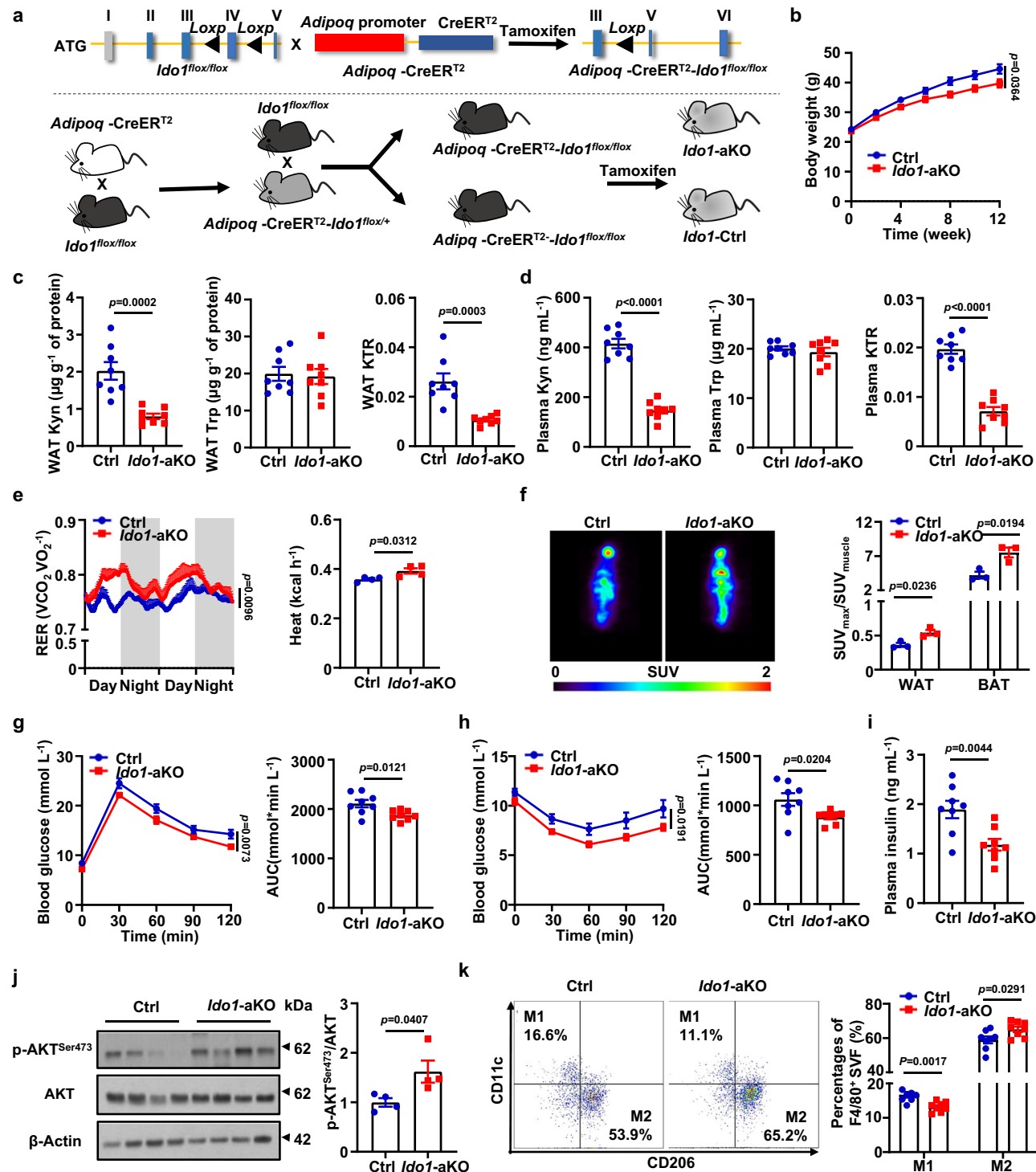

is consistent with the results obtained from the *Ido1*$^{-/-}$ mice (Supplementary Fig. 2q). Collectively, our data support that obesity is coupled with IDO1 overexpression in mature adipocytes, thereby leading to the excessive production of Kyn. Therefore, specific depletion of *Ido1* in adipocytes protected mice from HFD-induced obesity by limiting Kyn production.

**Kyn mediates metabolic disorder and insulin resistance via the AhR/STAT3/IL-6 axis.** The above findings suggest that Kyn likely exerts its impact on adipocytes instead of macrophages (Fig. 2 and Supplementary Fig. 5) in individuals with obesity. Since the *Ido1*-aKO mice exhibited a dramatically increased

lipolysis and decreased lipogenesis coupled with relieved adipocyte hypertrophy in the WAT, we assume that Kyn impairs lipid metabolism in adipocytes predisposing to the development of obesity upon sustained HFD challenge. To address this issue, 3T3L1-derived mature adipocytes were cultured under gradient concentrations of Kyn (0–100 μg mL$^{-1}$). It was noted that Kyn dose-dependently promoted lipogenesis as illustrated by the increased lipid deposition (Fig. 4a, indicated by Oil Red staining). Consistently, Kyn stimulated a significant up-regulation of lipogenic genes (*C/ebpα*, *Pparγ*, *Fabp4*) (Fig. 4b). Western blot analysis further confirmed that Kyn decreased the level of p-ACC$^{Ser79}$ in favor of lipogenesis, but reduced the level of

**Fig. 3 Ido1 deficiency in adipocytes renders the mice with decreased Kyn level and resistance to obesity. a** Based on the CRISPR-Cas9 system, the *Ido1*flox/flox mice were generated by inserting two loxP sequences in the same direction into the intron flanking exon 4 of IDO1, which would generate a stop codon in exon 3 to produce a nonfunctional IDO1 protein after Cre-mediated gene deletion. The *Ido1*flox/flox mice were crossed with the Adipoq-CreER$^{T2}$ mice to get the Adipq-CreER$^{T2}$-*Ido1*flox/flox mice. After intraperitoneally injected with tamoxifen (75 mg/kg/d) for 5 days, the *Ido1*-aKO mice and their littermates were employed for following studies. **b** Body weights of Ctrl and *Ido1*-aKO mice fed with HFD for 12 weeks (*n* = 8). Kyn, Trp concentration and KTR in eWAT (**c**) and plasma (**d**) from Ctrl and *Ido1*-aKO mice fed with HFD for 12 weeks (*n* = 8). **e** Respiratory exchange ratio (left) and heat production (right) of HFD-fed Ctrl and *Ido1*-aKO mice (*n* = 4). **f** Representative images of $^{18}$F-FDG micro-PET/CT (left) and quantification (right) of tissue FDG uptake in Ctrl and *Ido1*-aKO mice fed with HFD for 12 weeks (SUV$_{muscle}$ was used as a standardized value) (*n* = 3). GTT (**g**) and ITT (**h**) of Ctrl and *Ido1*-aKO mice fed with HFD for 12 weeks (left) and areas under curves (AUC) (right) (*n* = 8). **i** Plasma insulin levels of the Ctrl and *Ido1*-aKO mice (*n* = 8). **j** Western blot analysis of p-AKT$^{Ser473}$ and AKT, in eWAT from Ctrl and *Ido1*-aKO mice fed with 12-week HFD (*n* = 4). **k** Flow cytometry analysis of macrophage subsets in the eWAT of Ctrl mice and *Ido1*-aKO mice (*n* = 8). Data were represented as mean ± SEM. RER in **e** was analyzed by the moving average. Student's *t* test was used for statistical analysis. Statistical significance was assessed by two-way ANOVA followed with Bonferroni's multiple comparisons test (**b**, **e**, **g** and **h**), two-sided Student's *t* test (**c**, **d**, **f**, **i**–**k**) and significant differences were indicated with *p* values. Source data are provided in the Source Data file.

p-HSL$^{Ser660}$ against lipolysis (Fig. 4c). Altered lipid metabolism then impaired insulin sensitivity featured by the reduced p-AKT$^{Ser473}$ levels (Fig. 4c).

Given that AhR is a canonical receptor for Kyn[30,31], we, therefore, next examined AhR expression in Kyn stimulated adipocytes. As expected, Kyn dose dependently induced AhR expression (Fig. 4d, upper panel). Moreover, a significant decline of AhR expression was observed in mature adipocytes originated from HFD induced *Ido1*-aKO mice (Supplementary Fig. 7d), as well as an increase of AhR in mature adipocytes originated from Kyn-treated control mice (Supplementary Fig. 7e), indicating an essential role for Kyn in the induction of AhR expression. We next screened a number of signaling molecules relevant to lipogenesis in Kyn stimulated adipocytes, and the signal transducer and activator of the transcription 3 (STAT3) was characterized with increased expression both at total protein and phosphorylated levels following Kyn stimulation (Fig. 4d, middle panel). Similar tendency was also observed in mature adipocytes originated from Kyn-treated control mice (Supplementary Fig. 7e), which was blunted by the induction of *Ido1* deficiency (Supplementary Fig. 7d). In silico analysis was then conducted, and an AhR binding site was characterized within the *STAT3* promoter region (from −341 bp to −334 bp, transcriptional start site was defined as +1 bp, Fig. 4e). Chromatin immunoprecipitation (ChIP) assay was then employed and validated that AhR indeed selectively bound to the *STAT3* promoter (Fig. 4f). To confirm this observation, luciferase reporter assays were conducted and confirmed that AhR transcribed *STAT3* expression upon binding to its promoter (Fig. 4g, h), while disrupting its binding motif (the mutant plasmid) attenuated the transcriptional activity (Fig. 4g, h).

Since hepatic steatosis (Supplementary Fig. 6b) was noted other than the abnormalities of adipocytes (Supplementary Fig. 6c), the above results provided evidence that *Ido1* deficiency in adipocytes likely elicits a systemic impairment on metabolism. We thus conducted comparative analysis of inflammatory cytokines between HFD fed *Ido1*-aKO mice and WT mice, as STAT3 potently transcribes inflammatory cytokines relevant to systemic metabolism[32]. IL-6 was identified as the most significant one that was down-regulated in *Ido1*-deficient adipocytes (Supplementary Fig. 7a), which was further confirmed by ELISA analyses of plasma and WAT samples (Supplementary Fig. 7b, c). These results prompted us to check the expression of IL-6, a transcriptional target of STAT3[32], and indeed, a significant increase of IL-6 was detected in the culture supernatants of Kyn-treated adipocytes (Fig. 4i). Consistently, the plasma IL-6 level was increased in HFD-induced mice as compared to that of NCD-fed mice, which was even higher in HFD + Kyn group (Supplementary Fig. 7f). Similar tendency was also observed in the WAT IL-6 levels (Supplementary Fig. 7g).

To further confirm the above results, a rescue experiment was conducted in adipocytes. For this purpose, adipocytes were stimulated with Kyn in the presence of StemRegenin 1 (SR1), an AhR inhibitor or Stattic, a p-STAT3 inhibitor[33]. Addition of SR1 significantly alleviated Kyn-induced insulin resistance as characterized by the increased p-AKT$^{Ser473}$ along with decreased STAT3, p-STAT3$^{Tyr705}$ (Fig. 4j) and reduced IL-6 secretion (Fig. 4k). Furthermore, addition of Stattic rescued the phenotype of adipocytes from Kyn stimulation (Fig. 4j, k) while without affecting AhR expression (Fig. 4j), supporting that STAT3 is downstream of AhR. Luciferase reporter assays were also employed to confirm the above observations. Indeed, SR1 exhibited a comparative effect on inhibiting IL-6 expression as Stattic (Supplementary Fig. 7h).

Finally, we sought to address that the effect of Kyn is mediated by IL-6. Due to the originality of IL-6 might affect the physiological metabolic response[34,35], we did not use the exogenous IL-6 to stimulate adipocyte. Instead, Tocilizumab (TCZ), an IL-6 receptor blocking antibody, was employed to block the effect of adipocyte-derived endogenous IL-6 in the rescue experiments. Once IL-6 signaling was blocked by TCZ, Kyn-induced lipid deposition was almost completely abolished (Fig. 5a). Western blot results also indicated an enhanced lipolysis and insulin sensitivity along with an attenuated lipogenesis in TCZ-treated cells (Fig. 5b). Consistently, administration of TCZ significantly attenuated Kyn-induced increase of body weight (Fig. 5c), along with improved glucose tolerance (Fig. 5d) and insulin sensitivity (Fig. 5e, g), coupled with reduced adipocyte hypertrophy (Fig. 5f). Blockade of IL-6 signaling by TCZ also rendered animals with enhanced lipolysis and repressed lipogenesis (Fig. 5g) following Kyn challenge.

Another critical question is whether IFN-γ is implicated in the induction of *Ido1* expression, which then initiates the AhR/STAT3/IL-6 signaling within the adipocytes. Indeed, an increased IFN-γ expression was observed in the WAT of HFD-fed mice (Supplementary Fig. 7k). This observation promoted us to treat adipocytes with exogenous IFN-γ. In line with our expectation, IFN-γ potently stimulated IDO1 overexpression in adipocytes along with enhanced AhR and STAT3 signaling (Supplementary Fig. 7l). The enhanced downstream *Il-6* expression was also confirmed by RT-PCR (Supplementary Fig. 7m) and ELSIA analysis of culture supernatants (Supplementary Fig. Sn).

**Kyn promotes obesity and insulin resistance depending on AhR.** The next key question is whether Kyn promoting obesity and insulin resistance depends on AhR activation in animals. To this end, the adipocyte-specific *Ahr* knockout mice (the *Adipoq*-CreER$^{T2}$-*Ahr*flox/flox mice, hereinafter defined as *Ahr*-aKO mice) were generated by crossing the *Ahr*flox/flox mice with the *Adipoq*-CreER$^{T2}$ mice, and their littermates (the *Adipoq*-CreER$^{T2−}$-

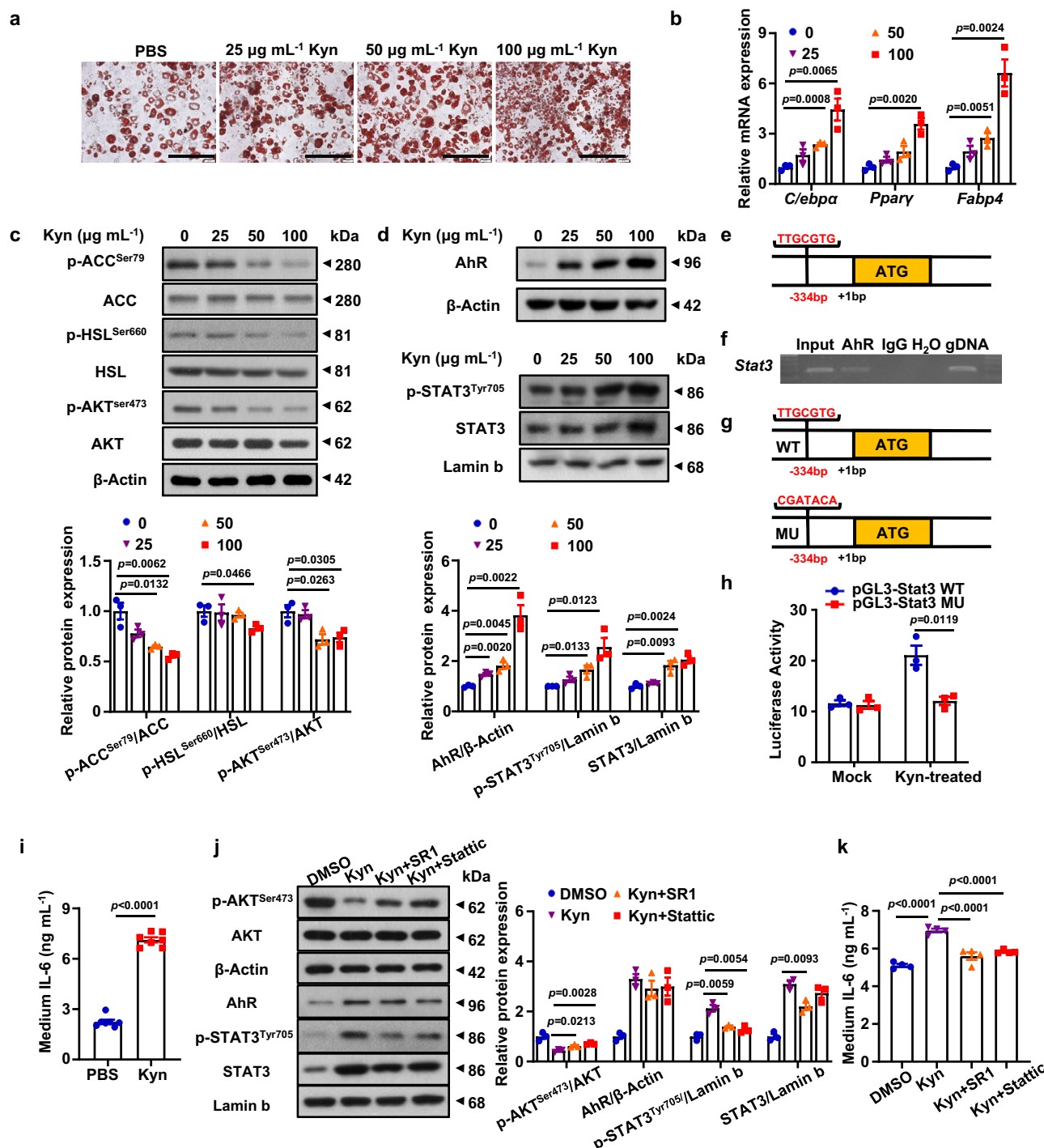

*Ahr*^flox/flox mice, hereinafter defined as Ctrl mice) were employed as controls (Fig. 6a). In line with our expectation, the *Ahr*-aKO mice displayed attenuated increase of body weight following a 12-week of HFD induction (Fig. 6b) along with improved glucose tolerance (Fig. 6c) and insulin sensitivity (Fig. 6d), and lower plasma insulin levels (Fig. 6e). Consistently, metabolic assays revealed an elevated carbohydrate utilization with increased energy expenditure in the *Ahr*-aKO mice as evidenced by the higher RER and heat production (Fig. 6f), while no significant difference was observed in terms of cumulative food intake between the KO and WT mice (Supplementary Fig. 8a). Furthermore, the *Ahr*-aKO mice displayed reduced WAT mass (Fig. 6g, upper panel) along with decreased adipocyte

hypertrophy (Fig. 6g, lower and side panel), enhanced lipolysis and attenuated lipogenesis (Supplementary Fig. 8b). Similarly, *Ahr* deficiency protected the mice from hepatic steatosis (Supplementary Fig. 8c).

It is noteworthy that Kyn concentrations in the WAT and plasma from *Ahr*-aKO mice were comparative to the Ctrl mice (Fig. 6h), indicating that the protection observed in the *Ahr*-aKO mice was not caused by the reduction of Kyn. However, both total STAT3 and p-STAT3^Tyr705 levels were markedly decreased in adipocytes from *Ahr*-aKO mice (Fig. 6i), which were further confirmed by RT-qPCR (Fig. 6j). Similarly, IL-6 was featured by the decrease at both mRNA (Fig. 6j) and protein (Fig. 6k) levels. As a result, the *Ahr*-aKO mice displayed increased insulin

**Fig. 4 Kyn mediates metabolic disorder and insulin resistance in adipocytes via the AhR/STAT3/IL-6 axis. a** 3T3-L1-derived mature adipocytes were stimulated with gradient concentrations of Kyn (0, 25, 50, 100 µg mL$^{-1}$), representatively. Oil Red staining of lipid droplets within the adipocytes was recorded. The experiments were repeated independently three times. Scale bar, 100 µm. **b** Transcriptional levels of lipogenesis associated genes (*C/ebpα*, *PPARγ*, *Fabp4*) in the above adipocytes (n = 3). **c** After the treatment of Kyn (0, 25, 50, 100 µg mL$^{-1}$) and palmitate (0.25 mmol L$^{-1}$, PA, Sigma), 100 nM insulin was added into the medium for 15 min before collecting the adipocytes. Western blot was conducted to detect p-ACC$^{Ser79}$/ACC, p-HSL$^{Ser660}$/HSL, p-AKT$^{Ser473}$/AKT in the adipocytes. The experiments were performed independently three times, and the quantitative analysis was shown as a bar graph in the bottom panel (n = 3). **d** AhR, p-STAT3$^{Tyr705}$ and STAT3 expression levels in above adipocytes were determined by Western blot. β-Actin or Lamin b was used as internal controls. The quantitative analysis was shown as a bar graph in the bottom panel (n = 3 independent experiments). **e** The predicted AhR binding site within the *Stat3* promoter. **f** ChIP results for the analysis of AhR binding activity to the *Stat3* promoter. gDNA: guide DNA. **g**, **h** Relative luciferase activity in 3T3-L1. MU: mutant. The experiments were repeated independently three times (n = 3). **i** ELISA analysis of IL-6 in the culture supernatants of adipocytes following Kyn (100 µg mL$^{-1}$) exposure for 48 h (n = 7). **j** Western blot analysis of p-AKT$^{Ser473}$/AKT, p-STAT3$^{Tyr705}$/STAT3 and AhR in mature adipocytes treated with SR1 or Stattic (n = 3 independent experiments). **k** ELISA analysis of IL-6 in the culture supernatants of adipocytes treated with SR1 or Stattic. The experiments were repeated independently four times (n = 4). Data were represented as mean ± SEM. Statistical significance was assessed by one-way ANCOVA (**b–d**, **j** and **k**), two-sided Student's *t* test (**i**), two-way ANCOVA (**h**) and significant differences were indicated with *p* values. Source data are provided in the Source Data file.

sensitivity evidenced by the increased p-AKT$^{Ser473}$ (Fig. 6l). Moreover, administration of exogenous Kyn failed to induce the up-regulation of STAT3 and IL-6 in the WAT from *Ahr*-aKO mice (Supplementary Fig. 8d), further reinforcing that Kyn exerts its impact on adipocytes in an AhR dependent manner.

**Vit-B6 confers protection by facilitating Kyn catabolism.** Finally, we sought to translate the above discoveries into a feasible therapeutic strategy. Given that pyridoxal 5'-phosphate (PLP), a biologically active form of vitamin B6 (hereinafter defined as Vit-B6), is a well-known cofactor of the key enzymes (including Kyn aminotransferases and kynureninase) for Kyn catabolism[13], we first checked PLP concentrations using aforementioned serum samples collected from subjects that were overweight or affected by obesity. It was noted that the subjects that were overweight, and particularly for subjects with obesity, were featured by the Vit-B6 deficiency as compared to the lean controls (Fig. 7a). Similar results were also obtained in HFD induced obese mice both in the plasma and WAT samples (Fig. 7b, c). In contrast, increased PLP was observed in the liver and skeletal muscle (Supplementary Fig. 9a, b), which might be a compensatory manner. Next, we stimulated adipocytes with Kyn in the presence of PLP. Remarkably, addition of PLP almost completely abolished Kyn-induced AhR expression coupled with a restoration of STAT3 and p-STAT3 levels (Fig. 7d and Supplementary Fig. 9c). Attenuated IL-6 secretion (Fig. 7e) and increased p-AKT (Supplementary Fig. 9d) were also observed. Together, those data imply that PLP promotes Kyn catabolism, thereby blocking the AhR/STAT3/IL-6 signaling and improving insulin sensitivity.

Next, Vit-B6 was supplemented in the drinking water to those mice following 4-week of HFD challenge, and the mice were challenged by HFD for another 8 weeks. Consistently, the mice supplemented with Vit-B6 exhibited significantly lower body weight (Fig. 7f) along with improved glucose tolerance (Fig. 7g) and insulin sensitivity (Fig. 7h). To validate that the protection was attributed to the reduction of Kyn caused by PLP, we measured Kyn and PLP concentrations in the WAT and plasma from above mice, respectively. Indeed, the mice supplemented with Vit-B6 displayed significantly higher levels of PLP both in the WAT and plasma samples (Fig. 7i), while a striking reduction of Kyn levels was noted in the WAT and plasma (Fig. 7j). RT-qPCR analysis revealed repressed *Ahr*, *Stat3* and *Il-6* expression in the WAT (Fig. 7k), which was further confirmed by Western blotting (Fig. 7l). In line with these results, the Vit-B6 supplemented mice manifested improved insulin sensitivity (Fig. 7m) along with declined adipocyte hypertrophy (Supplementary Fig. 9e). Collectively, Vit-B6 supplementation can effectively prevent Kyn accumulation, thereby protecting mice

from HFD-induced obesity, which could be a promising therapeutic strategy against obesity in clinical settings.

**Discussion**
Altered glucose and lipid metabolism are implicated in the pathogenesis of obesity, while aberrant amino acid metabolism in individuals with obesity tends to be overlooked[6]. Previous studies suggested feasible evidence that the essential amino acid tryptophan is preferentially catabolized through the kynurenine pathway (KP) in subjects with obesity, leading to an increase of circulating Kyn[9,26]. However, the exact seminary of the incremental Kyn remains to be elucidated. Herein, we demonstrated that mature adipocytes from subjects with obesity are featured by the overexpression of IDO1, thereby mediating tryptophan catabolism to produce excessive Kyn, which in turn exacerbates obesity and insulin resistance. Mechanistically, Kyn activates AhR, which then transcribes STAT3 expression to enhance IL-6 production. In this case, altered IDO1 expression leads to Kyn accumulation, which enhances AhR/STAT3/IL-6 signaling in adipocytes to mediate obesity and insulin resistance. Moreover, supplementation of Vit-B6, a cofactor for Kyn catabolism, significantly reduced Kyn levels in the WAT and plasma, thereby protecting mice from HFD-induced obesity and insulin resistance. Therefore, elimination of accumulated Kyn could be viable strategy against obesity.

In general, more than 95% of dietary Trp is catabolized through the kynurenine pathway[36]. Three distinct dioxygenase enzymes, the tryptophan 2,3-dioxygenase (TDO), IDO1 and IDO2 are responsible for the catabolism of Trp to Kyn. TDO is predominantly expressed in the liver, skeletal muscle, placenta, and brain[37]. In contrast, IDO1 manifests a ubiquitous expression pattern, and IDO2 is a paralogue of IDO1, and is constitutively expressed in the liver, kidney tubules, spermatozoa, and dendritic cells[38]. Moreover, the catalytic ability of IDO1 (*Km*, 20.9 µmol L$^{-1}$) is far beyond that of IDO2 (*Km*, 6890 µmol L$^{-1}$)[39]. Additionally, IDO1 is more relevant to the pathological conditions, particularly in individuals with obesity[9]. Although IDO1 overexpression has been noted in the WAT, small intestine and colon from subjects with obesity, less effort has been devoted to explore the exact cell type contributing to the superfluous Kyn production[9,26]. To address this question, mature adipocytes and SVF were separated from WAT for analysis of IDO1 expression. IDO1 overexpression was only detected in mature adipocytes (Fig. 2c, d), which was further confirmed in human samples (Fig. 2e, f). Studies in adipocyte-specific *Ido1* knockout mice provided convincing evidence that mature adipocytes serve as the major source of circulating Kyn in obese mice (Fig. 3). Specifically, depletion of *Ido1* in adipocytes dramatically reduced

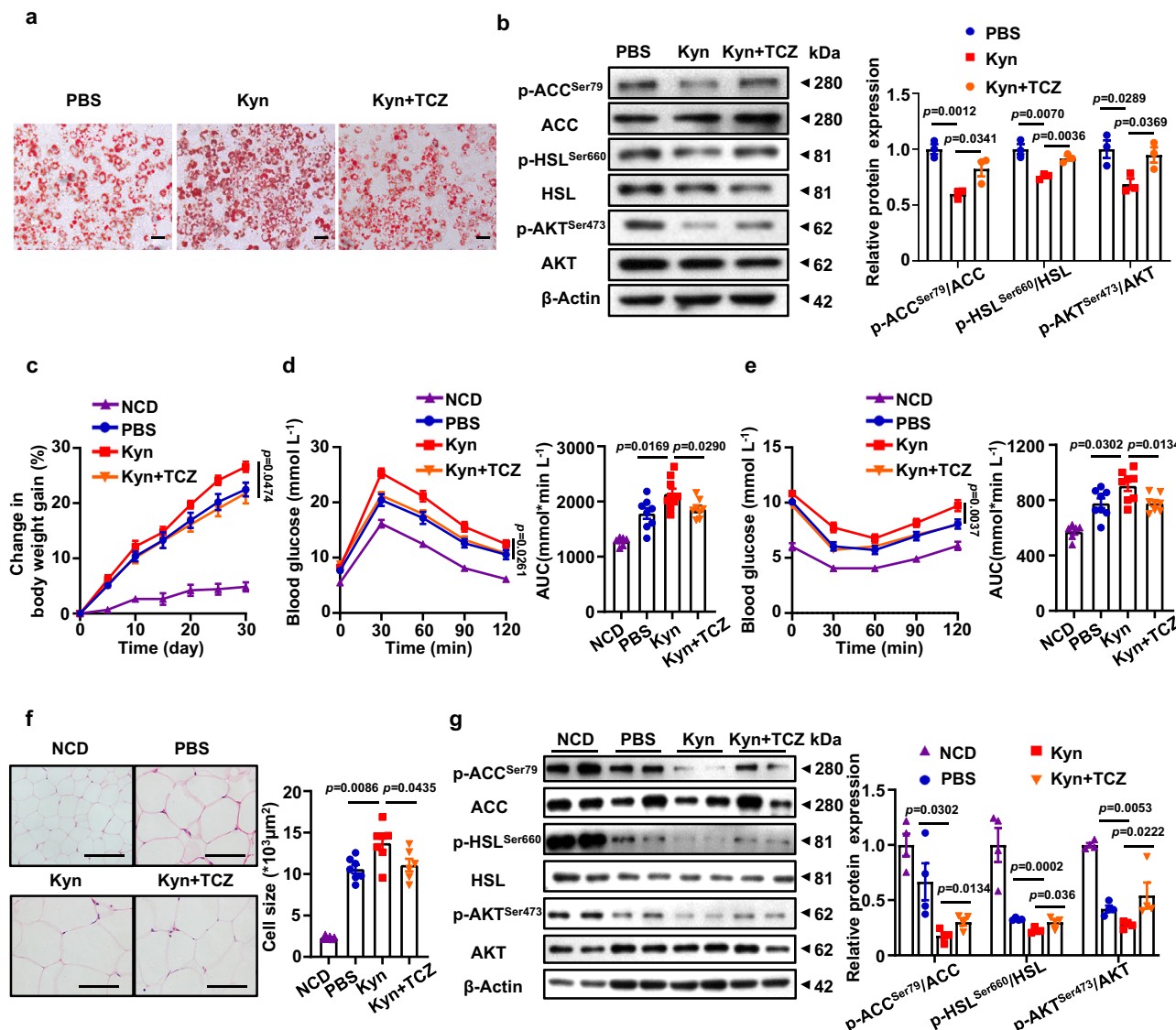

**Fig. 5 The effect of Kyn is mediated by IL-6. a** Kyn (100 μg mL$^{-1}$) and TCZ (100 μg mL$^{-1}$) were added to stimulate 3T3-L1 derived adipocytes. Representative Oil Red staining images of the mature adipocytes. The experiments were performed independently three times. Scale bar, 100 μm. **b** After the treatment of Kyn (100 μg mL$^{-1}$) with or without TCZ (100 μg mL$^{-1}$), 100 nM insulin was added into the medium for 15 min before collecting the adipocytes. Western blot was conducted to detect p-ACC$^{Ser79}$/ACC, p-HSL$^{Ser660}$/HSL, p-AKT$^{Ser473}$/AKT in adipocytes (n = 3 independent experiments). **c** Body weight change of WT mice fed with 8-week NCD (n = 6) or HFD treated with PBS (n = 8) or Kyn (20 mg kg$^{-1}$) (n = 10) or TCZ (5 mg kg$^{-1}$) (n = 8) for another month, respectively. GTT (**d**) and ITT (**e**) of WT mice fed with 8-week NCD (n = 6) or HFD treated with PBS (n = 8) or Kyn (20 mg kg$^{-1}$, n = 9 for IGTT and n = 8 for ITT, respectively) or TCZ (5 mg kg$^{-1}$, n = 8) for another month. **f** Representative H&E staining images of eWAT originated from 8 weeks NCD and HFD fed mice following PBS, Kyn or TCZ treatment for one month (n = 6). Scale bar, 100 μm. **g** Western blot analysis of p-ACC$^{Ser79}$/ACC, p-HSL$^{Ser660}$/HSL, p-AKT$^{Ser473}$/AKT in eWAT from NCD, HFD, HFD+Kyn-treated mice and HFD+Kyn+TCZ-treated mice (n = 4). Statistical significance was assessed by one-way ANOVA (**b**, **f** and **g**) or two-way ANOVA followed with Bonferroni's multiple comparisons test (**c**, **d** and **e**) and significant differences were indicated with p values. Source data are provided in the Source Data file.

circulating Kyn in HFD-fed mice (Fig. 3d), which was even lower than that of NCD-fed WT mice (Fig. 1d). The discrepancy of plasma Kyn between HFD-fed KO mice and NCD-fed WT mice indicates that IDO1 in adipocytes not only mediates altered Kyn production in subjects with obesity, but also involves in the maintenance of Kyn homeostasis in healthy individuals. Unexpectedly, no significant difference in terms of Trp between control mice and *Ido1*-aKO mice was observed (Fig. 3c, d), which could be attributed to the enhanced catabolism of Trp through serotonin/melatonin pathway[11].

Unlike the observations noted in the plasma and WAT, Kyn concentrations in the liver and skeletal muscle did not show a

significant difference between HFD and NCD-fed mice (Fig. 2a). As mentioned earlier, this discrepancy is likely caused by the expression differences of isozymes for Trp catabolism in these organs[37]. In liver and skeletal muscle, TDO is predominantly expressed, which is in charge for maintaining the basal level of Kyn. Compared to TDO, IDO1 is inducible and more sensitive to the pathological insults. For example, during the course of HFD challenge, IDO1 was dramatically overexpressed in the WAT (Fig. 2b) accompanied by increased accumulation of Kyn (Fig. 2a), while no perceptible difference was noted in the liver and skeletal muscle between HFD and NCD-fed mice (Fig. 2a, b). The increase of Kyn in WAT during obesity could also be

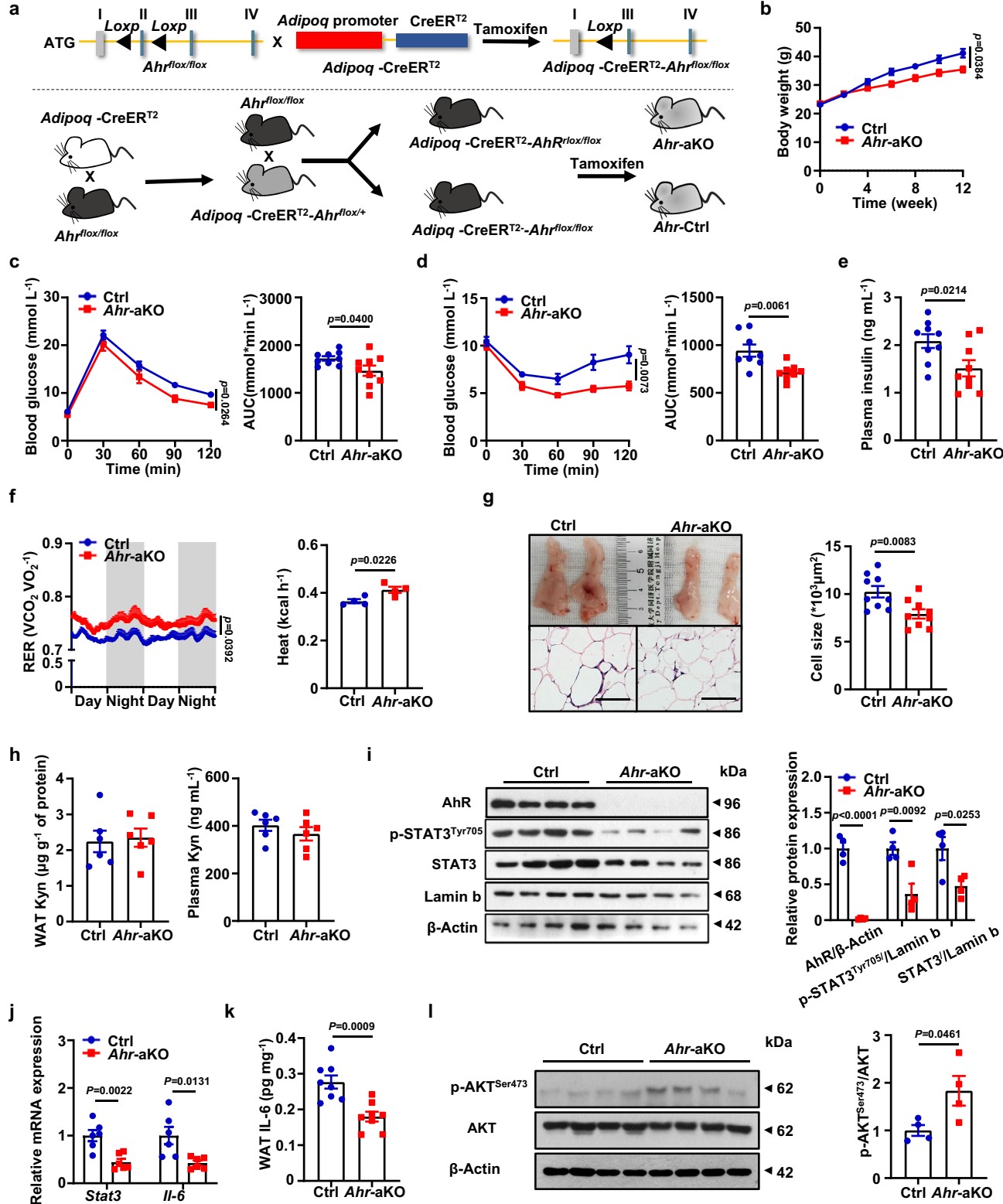

logically explained by the discrepancy noted in terms of PLP levels (Fig. 7c), a key coenzyme to catalyze Kyn catabolism. In sharp contrast, the PLP level in the liver or skeletal muscle even exhibited a compensatory elevation in obesity (Supplementary Fig. 9a, b). Collectively, the overexpressed IDO1 along with the decreased PLP synergistically leads to an elevation of Kyn in WAT during obesity, subsequently boosting the plasma Kyn.

Generally, Kyn and IDO1 are considered as immunosuppressive factors involving in the repression of effector immune cells and induction of tolerogenic immune cells[14]. For instance, it was reported that IDO1 overexpression could promote macrophage M2 program, while inhibition of IDO1 would induce M1 polarization[40]. In contrast to these studies, we observed a dramatic increase of Kyn and IDO1 in subjects with obesity (Figs. 1

**Fig. 6 Kyn promotes obesity and insulin resistance depending on AhR. a** The *Ahr*[flox/flox] mice were crossed with the *Adipoq*-CreER[T2] mice to generate the *Adipoq*-CreER[T2]-*Ahr*[flox/flox] mice. After intraperitoneally injected tamoxifen (75 mg kg⁻¹ d⁻¹) for 5 days, the *Ahr*-aKO mice and their littermates were obtained. **b** Body weight for the Ctrl and *Ahr*-aKO mice following 12 weeks of HFD induction ($n = 9$). GTT (**c**) and ITT (**d**) of the Ctrl and *Ahr*-aKO mice after 12-week HFD ($n = 9$). **e** Plasma insulin levels of the Ctrl and *Ahr*-aKO mice fed with HFD for 12 weeks ($n = 9$). **f** RER and heat production of the mice ($n = 4$). **g** Images of eWAT and representative H&E staining images of eWAT originated from the Ctrl and *Ahr*-aKO mice (left), and the statistical analysis of adipocyte sizes (right) ($n = 9$). Scale bar, 100 μm. **h** Kyn concentration in eWAT (left) and plasma (right) originated from the Ctrl and *Ahr*-aKO mice ($n = 8$). **i** Western blot of p-STAT3[Tyr705]/STAT3 and AhR in adipocytes from the Ctrl and *Ahr*-aKO mice ($n = 3$). **j** *Stat3* and *Il-6* in eWAT from the Ctrl and *Ahr*-aKO mice were measured by RT-qPCR ($n = 6$). **k** ELISA analysis of IL-6 concentration in eWAT ($n = 7$). **l** Western blot results of p-AKT[Ser473] and AKT in eWAT ($n = 4$). Data were represented as mean ± SEM. RER in **f**, was analyzed by moving average. Statistical significance was assessed by two-way ANOVA followed with Bonferroni's multiple comparisons test (**b**, **c**, **d** and **f**), two-sided Student's *t* test (**e**, **g–l**) and significant differences were indicated with *p* values. Source data are provided in the Source Data file.

and 2). Since IDO1 can be induced by proinflammatory factors, such as IFN-γ (Supplementary Fig. 7k) and TNF-α[41], it is reasonable to assume that Kyn and IDO1 are upregulated by the obesity-induced low-grade chronic inflammation in a compensated manner. Nevertheless, administration of exogenous Kyn significantly exacerbated HFD-induced obesity and insulin resistance (Fig. 1f–i), and whole-body knockout of *Ido1* restored circulating Kyn levels and protected mice from obesity (Fig. 1j–m and Supplementary Fig. 2k–q), which were opposite to the assumption. In line with these results, studies in the *Ido1*-lKO mice demonstrated a dispensable role of IDO1 for macrophages in individuals with obesity (Supplementary Fig. 5), which in fact was consolidated by a bone marrow transplantation study[9].

The above described paradox prompted us to exclude the possibility that Kyn exacerbates obesity by impacting the function of adipose tissue macrophages (ATMs). In particular, IL-6 can also be produced by M1 macrophages. To exlude the possibility that higher circulating IL-6 in obese mice was also attributed by ATMs other than adipocytes, we firstly induced M1 macrophages using bone-marrow derived macrophages (BMDM). The polarized M1 macrophages were stimulated with different concentrations of Kyn. Unexpectedly, at the concentration of 50 μg mL⁻¹ and 100 μg mL⁻¹ Kyn, the production of IL-6 by M1 macrophages tended to be reduced (Supplementary Fig. 7i, j), which was opposite to what we observed in adipocytes. This interesting phenomenon was also observed by Wang and colleagues, in which the production of IL-6 was blunted in LPS-induced THP-1 following Kyn stimulation[42]. Therefore, it is likely that IL-6 is predominantly produced by adipocytes in individuals with obesity.

The exact role of IL-6 signaling in obesity and insulin resistance is yet to be fully elucidated, even controversial results had been reported. However, recent studies have demonstrated that the controversial effect of IL-6 signaling on obesity and insulin resistance might be attributed to its originality[34]. It is beieved that IL-6 secreted by adipocytes during obesity would promote chronic inflammation and exacerbate metabolic syndrome[34,35]. To verify the effect of Kyn-induced IL-6 on regulating the synthesis and decomposition of fat in adipocytes, we conducted rescue experiments. Due to that the exogenous IL-6 might affect the physiological metabolic response[34,35], Tocilizumab (TCZ), a IL-6 receptor blocking antibody, was applied to block the effect of adipocyte-derived endogenous IL-6 in the rescue experiments. Indeed, we discovered that TCZ ameliorated the impact caused by Kyn-induced IL-6 (Fig. 5), indicating that the effect of Kyn on obesity and insulin resistance is mediated by IL-6.

It is worthy of note that our current studies employed a larger cohort confirmed a significant increase of Kyn in patients and mice with obesity (Fig. 1a–d). As discussed above, we also demonstrated convincing evidence that obesity is coupled with PLP deficiency which catalyzes Kyn to Kyna[13], both in humans and mice (Fig. 7a–c). Agudelo et al. recently also reported that exercised skeletal muscle can increase the catabolism of Kyn to

Kyna[43], thereby increasing energy expenditure by activating Gpr35 and further improving energy metabolism in mice fed with high-fat diet[44]. Their data in fact are consistent with our results, in which alleviation of Kyn accumulation would improve metabolic homeostasis. Additional studies also addressed a consistent role of Kyn in individuals with obesity[9,10].

Since adipocytes are characterized to be the predominant cell type responsible for the excessive Kyn production in individuals with obesity, it would be logical to assume that Kyn impacts the metabolic homeostasis and functions of adipocytes directly. Indeed, Kyn stimulated lipid deposition and impaired insulin sensitivity in adipocytes (Fig. 4a–c). Further mechanistic studies revealed that the impacts of Kyn on adipocytes depended on the expression of AhR and the activation of its downstream STAT3/IL6 signaling pathway (Fig. 4d–k). Therefore, specific depletion of AhR in adipocytes robustly abolished the effect of Kyn (Figs. 4k and 6). As aforementioned, those data were contradictory to the previously published data, in which Kyn and IDO1 play an immunosuppressive role in terms of chronic inflammation in obesity[17]. The possibility causes this discrepancy probably due to the increase of intestinal IDO1 activity[9]. In this case, Trp metabolism in the intestine could be shifted towards Kyn production, which would diminish the indole derivatives, such as indole-3-acetic acid, and IL-22 produced by the gut microbiota, thereby enhancing chronic inflammation in individuals with obesity[9].

PLP, the active form of Vit-B6, serves as a cofactor of the key enzymes for Kyn catabolic processes[13,45]. We first demonstrated that subjects that were overweight or affected by obesity and HFD-induced mice were characterized by PLP deficiency (Fig. 7a–c). Of particular relevance, Vit-B6 supplementation significantly attenuated HFD-induced obesity and insulin resistance by declining Kyn accumulation (Fig. 7f–m). These data support that elimination of Kyn accumulation in adipocytes could be a viable approach to prevent obesity. Given the crucial role of adipocytes played in Kyn production, a targeted drug delivery strategy aimed to directionally catalyze Kyn catabolism in adipocytes would be a major focus in the future studies.

In summary, our studies highlighted a central role for mature adipocytes in altered Kyn production in individuals with obesity (Fig. 8). Mechanistic studies revealed that Kyn impairs lipid homeostasis and insulin sensitivity in adipocytes rather than directly impacts immune cells. Specifically, Kyn stimulates AhR expression to activate the AhR/Stat3/IL-6 signaling in adipocytes, by which it mediates a systemic effect on the development of obesity and insulin resistance. Therefore, targeting IDO1 for reducing Kyn production combined with Vit-B6 supplementation for promoting Kyn catabolism could be a viable strategy against obesity and insulin resistance in clinical settings.

## Methods

**Mouse models.** The *Ido1* knockout (*Ido1*⁻/⁻) mice (B6.129-*Ido1*[tm1Alm]/J; #005867), *Adipoq*-creER[T2] mice [C57BL/6-Tg (*Adipoq*-cre/ERT2)1Soff/J; #025124], *LyzM*-Cre mice (B6.129P2-*Lyz2*[tm1(cre)Ifo]/J; #004781) were purchased

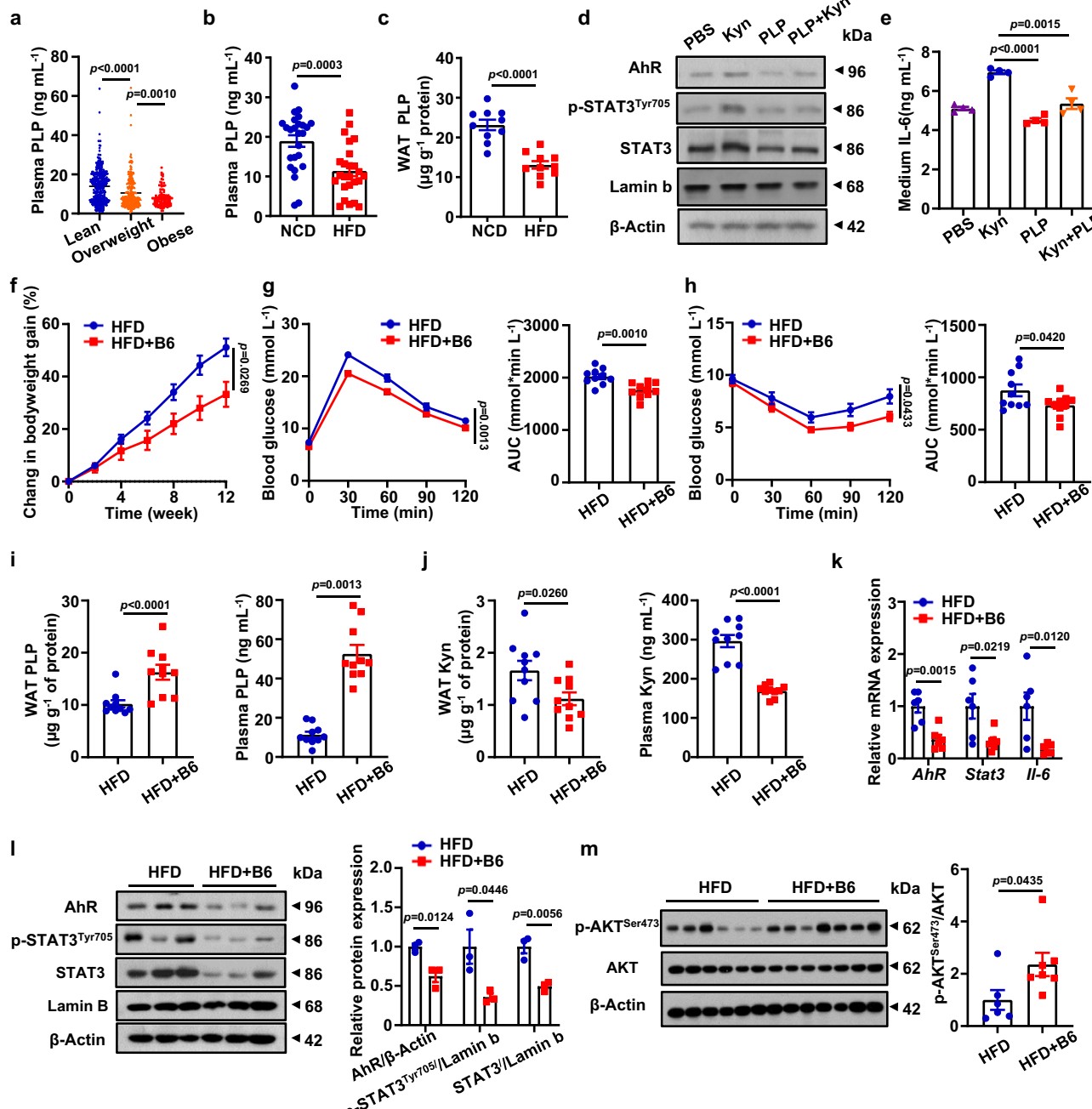

**Fig. 7 Vit-B6 confers protection by facilitating Kyn catabolism. a** Plasma PLP levels in the lean subjects ($18 \leq$ BMI $< 24$, $n = 354$), subjects with overweight ($24 \leq$ BMI $< 28$, $n = 268$) and subjects with obesity (BMI $\geq 28$, $n = 113$). **b** Plasma PLP levels in WT mice fed with a 12-week of NCD or HFD ($n = 25$). **c** eWAT PLP levels in WT mice fed with a 12-week of NCD or HFD ($n = 10$). **d** Western blot results for p-STAT3$^{Tyr705}$/STAT3 and AhR in mature adipocytes with indicated treatments ($n = 3$ independent experiments). **e** ELISA analysis of IL-6 in the culture supernatants of adipocytes with indicated treatments. **f** Bodyweight of HFD fed mice and HFD + B6 fed mice ($n = 8$). GTT (**g**) and ITT (**h**) measured in HFD fed mice and HFD + B6 fed mice ($n = 8$). PLP (**i**) and Kyn (**j**) levels in eWAT (left) and plasma (right) of mice in the HFD group and HFD + B6 group ($n = 8$). **k** RT-qPCR results for *Ahr*, *Stat3* and *Il-6* in eWAT of mice from HFD group and HFD + B6 group ($n = 6$). **l** Western blot results for p-STAT3$^{Tyr705}$/STAT3 and AhR in eWAT of mice from HFD group and HFD + B6 group. **m** Western blot for p-AKT$^{Ser473}$ and AKT in eWAT of mice from HFD group and HFD + B6 group ($n = 6$). Data were represented as mean ± SEM. Statistical significance was assessed by one-way ANOVA (**a**, **e**), two-sided Student's *t* test (**b**, **c**, **i**, **j**, **k**, **l** and **m**) and two-way ANOVA followed with Bonferroni's multiple comparisons test (**f**, **g** and **h**) and significant differences were indicated with *p* values. Source data are provided in the Source Data file.

from the Jackson Laboratory (Maine, USA). The *Ahr* floxed mice (*Ahr*$^{tm3.1Bra}$/J; #006203) was gifted by Sichuan University. The *Ido1* floxed mice were generated by the CRISPR-Cas9 system (Bioray Laboratories Inc., Shanghai, China). Relative genotyping results were shown in Supplementary Fig. 10d–h and knockout efficiencies of the mice were shown in Supplementary Fig. 10a–c and Fig. 6i. Adult wild type (WT) male C57BL/6 mice were purchased from the HFK Bioscience

(Beijing, China). The *Ido1* floxed mice were crossed with the *Adipoq*-creER$^{T2}$ mice or *LyzM*-Cre mice to generate the *Adipoq*-creER$^{T2+}$- *Ido1*$^{flox/flox}$ mice or *LyzM*-Cre$^+$-*Ido1*$^{flox/flox}$ mice, respectively. The *Ahr* floxed mice were crossed with the *Adipoq*-creER$^{T2}$ mice to generate the *Adipoq*-creER$^{T2+}$- *Ahr*$^{flox/flox}$ mice. The *Adipoq*-creER$^{T2+}$- *Ido1*$^{flox/flox}$ mice and *Adipoq*-creER$^{T2+}$- *Ahr*$^{flox/flox}$ mice as well as their littermate control mice were administered with intraperitoneal injection of

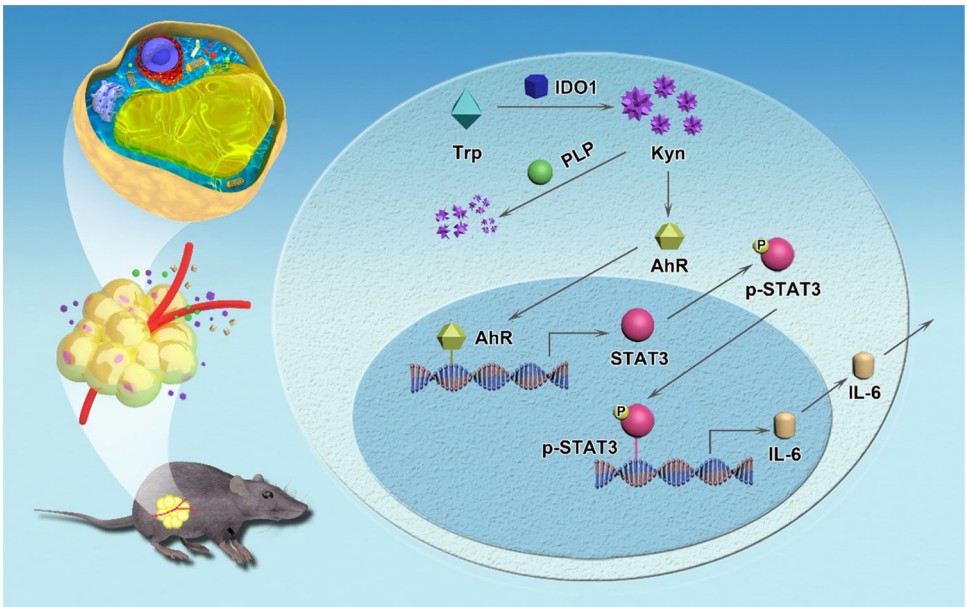

**Fig. 8 Scheme of adipocyte-derived kynurenine promoting obesity and insulin resistance.** Over nutrition induces overexpress IDO1 in adipocytes, which then produce copious amount of Kyn, thereby exacerbating insulin resistance and obesity via the AhR/STAT3/IL-6 axis, while PLP catalyze Kyn catabolism to efficiently prevent insulin resistance and obesity.

tamoxifen (75 mg kg$^{-1}$; Sigma) for 5 consecutive days for induction of *Ido1* deficiency. The male mice (8 weeks old) were housed individually and fed with high fat (60% kcal fat; D112492, Research Diet, Gardners, USA) or normal (9.4% kcal fat; 70032509, HFK Bioscience, Beijing, China) diet for 12 consecutive weeks, respectively.

For Kyn intervention studies, the male mice were fed with HFD for 8 weeks. After which, the mice were subcutaneously injected with L-Kyn (20 mg kg$^{-1}$ d$^{-1}$; Sigma-Aldrich Co., St. Louis, MO, USA) or PBS during the rest 4 weeks of HFD feeding. For Kyn rescue experiments, the male mice were pretreated with NCD or HFD for 8 weeks. The 8-week HFD mice were randomly divided into three groups: PBS, L-Kyn (20 mg kg$^{-1}$ d$^{-1}$), L-Kyn (20 mg kg$^{-1}$ d$^{-1}$) and TCZ (5 mg kg$^{-1}$ every other day, Roche Pharma, Schweiz, Ltd) for another 30 days. For vitamin B6 supplementation studies, WT male mice after a 4-week of HFD challenge were randomly divided into two groups: one was received HFD supplemented with vitamin B6 (2 g L$^{-1}$) for another 8 weeks, while the other was just maintained on HFD for the remaining 8 weeks.

All mice were housed in microisolator cages with free access to sterile acidified water and irradiated food for 12 h light-dark cycle and bred in the SPF animal facility of Tongji (Temperature: 20–24 °C; Humidity: 45–65%). All animal procedures were approved by the Tongji Hospital Animal Care (TJH-202001009) and Use Committee in line with the USA National Institutes of Health (NIH) guidelines.

**Human samples.** Fasting plasma samples were collected in Tongji Hospital from year 2016 to 2017. Subjects with renal or liver dysfunction were excluded from our study. Clinical characteristics of participants are summarized in Supplementary Table 1. Omental adipose tissue was obtained from patients (*n* = 22) undergone abdominal surgery for benign diseases. All patients were devoid of any evident systemic disease, any chronic infection or previous myocardial infarction. Omental adipose tissues were collected and mature adipocytes were isolated. Clinical characteristics of the participants are summarized in Supplementary Table 2. Consent form was obtained from each study subject. All human related studies were conducted in accordance with the NIH guidelines and were approved by the Institutional Review Board (IRB) of Tongji Hospital (TJ-IRB20160601, TJ-IRB20160602).

**Liquid Chromatography–tandem Mass Spectrometry (LC-MS) analysis.** LC-MS was employed to measure Trp, Kyn and PLP using the established techniques[46,47]. For Trp and Kyn measurement, 300 µL cold ddH$_2$O was added to 50–100 mg tissue samples and homogenized at 4 °C. The homogenized tissue/plasma was centrifuged at 12,000 × *g* for 10 min at 4 °C. 100 µL supernatant, 350 µL 1% formic acid acetonitrile and 50 µL internal standard (500 ng mL$^{-1}$ methyldopa) were aspirated to a new tube. After shaking for 30 s and centrifugation at 12,000 × *g* for 10 min at 4 °C, 300 µl supernatant was aspirated to a new tube and subsequently dried by nitrogen. Then, 100 µL 0.1% formic acid water was added to resuspend and centrifuged at 12,000 × *g* for 2 min at 4 °C. 90 µL supernatant was taken for LC-MS detection and analysis. For PLP measurement, the pretreatment of tissue

samples and plasma was the same as above. 100 µL prepared sample was aspirated with 100 µL internal standard (16% trichloroacetic acid: 0.01 mg mL$^{-1}$ 4-hydroxytoluene sulfobuturea was 100: 1), then the mixture was shaken for 30 s and centrifuged at 12,000 × *g* for 2 min at 4 °C. 100 µL supernatant was taken for LC-MS detection and analysis. 5 µL prepared sample was injected into an HPLC system) (Shimsdzu, Kyoto, Japan) equipped with Ultimate XB-C18 HPLC column (Welch Materials, Shanghai, China) at a column temperature of 40 °C. The tissue results were normalized by protein concentration. For Trp and Kyn measurement, solvent A was 0.1% formic acid water and solvent B was 0.1% formic acid acetonitrile. The gradient started at 2% B and increased to 80% B in 1.6 min, and then held at 80% B for 30 s. For PLP measurement, solvent A was 10 mM ammonium formate and solvent B was acetonitrile. The gradient started at 2% B and increased to 80% B in 2 min, and then held at 80% B for 50 s.

**Cell culture and treatment.** 3T3-L1 preadipocytes were maintained and differentiated into adipocytes as described previously[48]. Adipocytes were differentiated by treating confluent cells with induced medium [complete medium containing insulin (20 µg mL$^{-1}$, Sigma, #I6634), 3-isobutyl-1-methylxanthine (0.5 mM, Sigma, #I5879) and dexamethasone (1 µM, Sigma, #D1756)] for 48 h. And then the medium was changed to maintained medium (complete medium containing 20 µg mL$^{-1}$ insulin) for another 48 h. Generally, >95% of the cells began to acquire the adipocyte phenotype 3–4 days after initiating differentiation. For adipogenic study, exogenous Kyn (from 0 to 100 µg mL$^{-1}$) was added into the medium to check the impact of Kyn on adipogenesis. For insulin sensitivity analysis, mature adipocytes were undergone a 12 h of starvation (high-glucose DMEM with 2% FBS). The adipocytes were then treated with Kyn (100 µg mL$^{-1}$) for 12 h, followed by stimulating with palmitate (0.25 mmol L$^{-1}$, P0500, Sigma) for 48 h to induce insulin resistance. For rescue experiment, pyridoxal 5'phosphate (PLP, 100 µM, #P9255, Sigma)/Stattic (10 µM, #S7024, Selleck)/StemRegenin 1 (1 µM, #S2858, Selleck) was added into the medium for a 12 h of incubation. For IFN-γ stimulated experiment, mature adipocytes were conducted with IFN-γ (50 ng mL$^{-1}$) for 3 h. 100 nM insulin was added into the cultures for 15 min. The cells were harvested for the indicated studies.

BMDMs were generated as previously described with minor modifications[49,50]. Briefly, mouse femur-derived bone marrow cells were cultured in the presence of M-CSF (30 ng mL$^{-1}$, eBioscience, San Diego, CA, USA), and non-adherent cells were discarded at day 3 by changing to fresh media containing M-CSF. Adherent BMDMs were harvested at day 7 for experimental purpose. The prepared cells were pretreated with LPS (100 ng mL$^{-1}$) for 24 h, followed by stimulating with Kyn (50 µg mL$^{-1}$) for 4 h.

**Histological and morphological analysis.** Hematoxylin and Eosin (H&E) and Oil Red staining of epididymal adipose tissue and liver were carried out using the previously described techniques[48]. For Oil Red O staining, cultured adipocytes were washed in cold PBS gently and fixed in 10% neutral formalin for 30 min at room temperature. After washed by 60% isopropanol for 10 s, the cells were incubated with twice filtered 0.3% Oil Red-O solution for 10 min. Subsequently,

70% alcohol was used to rinse the cells and PBS stopped the reaction. Pictures were taken under a light microscope (OLYMPUS Upright microscope BX53, Olympus Corporation), and quantitative analysis was conducted using the Image J 1.46r software in a blinded fashion by two independent pathologists (National Institutes of Health, Wayne Rasband, USA).

**RT-qPCR analysis**. RT-qPCR was performed as previously reported[50]. RT-qPCR was then carried out using an ABI prism 7500 Sequence Detection System (Applied Biosystems, CA, USA). PCR amplifications were carried out at 95 °C for 1 min, followed by 40 cycles at 95 °C for 15 s, 60 °C for 1 min. Relative expression levels for each target gene were calculated using the $2^{-\Delta\Delta Ct}$ method. All primers were listed in Supplementary Table 3.

**Western blotting analysis**. Cells and tissues were lysed in RIPA buffer (50 mM Tris • HCl pH 7.4, 150 mM NaCl, 1% NP-40, 1% sodium deoxycholate, 0.1% SDS) containing 100X protease inhibitor cocktail (Roche, Indianapolis, IN, USA). Protein samples were separated with SDS-PAGE) and then transferred onto PVDF membrane (Bio-Rad). After blocking with 5% BSA, the membrane was incubated with primary antibodies for 16 h at 4 °C. Primary antibodies for phospho-AKT$^{Ser473}$ (4060s), AKT (4685s), phospho-HSL$^{Ser660}$ (4126s), phospho-ACC$^{Ser79}$ (3661s), ACC (3676s), phospho-GSK3β$^{Ser9}$ (9323T), GSK3β (12456s), phospho-STAT3$^{Tyr705}$ (9145s) and STAT3 (4904s) were obtained from the Cell Signaling Technology (Danvers, MA, USA), while antibodies against AhR (Sc-398877), Lamin B1 (Sc-20682), β-Actin (Sc-47778), PEPCK (Sc-32879), G6pase (Sc-25840) and HSL (Sc-25843) were ordered from the Santa Cruz Biotechnology (Santa Cruz, CA, USA). Primary antibody for IDO1 (122402) was originated from the Biolegend (San Diego, CA, USA). Primary antibody dilution was 1/1000. Then, HRP-conjugated secondary antibodies (1:5000) were incubated for 1 h at room temperature. β-Actin/Lamin B1 was used for normalization, and the intensity of each reactive band was analyzed using the Gelpro 32 software.

**Metabolic studies**. To analyze the metabolic index, the mice were individually placed in metabolic cages connected with a comprehensive laboratory animal monitoring systems (Columbus Instruments, Columbus, OH, USA). The mice were acclimatized to respiratory chambers for 48 h, followed by recording the real-time data of food intake, heat, oxygen consumption (VO$_2$), carbon dioxide production (VCO$_2$) and respiratory exchange ratio (RER). IGTT, ITT and insulin signaling assays were performed as described previously[23]. Blood glucose was measured by a glucometer (ACCU-CHEK, Roche, Germany). For IGTT, mice were fasted for 16 h before intraperitoneal injection of 1.5 g kg$^{-1}$ glucose (Sigma Co., St. Louis, MO, USA). Tail vein blood was collected at 0, 30, 60, 90 and 120 min to assay glucose concentration. Meanwhile, 20 μL 0 min blood was used to assay insulin concentration by mouse insulin ELISA kit (80-INSMSU-E01, ALPCO Diagnostics, USA). For ITT, mice were fasted for 4–6 h before intraperitoneal injection of 1 U kg$^{-1}$ insulin (Novolin R, Novo Nordisk Co., Bagsvaerd, Denmark) and colleted tail vein blood at 0, 30, 60, 90 and 120 min. For insulin signaling assays, mice were fasted overnight and then injected 5 U kg$^{-1}$ insulin intraperitoneally before sacrifice.

**Mature adipocytes and SVF isolation**. Fresh adipose tissue was cut into small pieces and transferred to a 50 mL centrifuge tube. Equal volume of digestion medium (2 mg mL$^{-1}$ Collagenase Type I containing 0.5 mg mL$^{-1}$ CaCl$_2$) was then added into the above prepared samples and mixed thoroughly. The mixture was digested at 37 °C with constant agitation at 150 rpm for 20–30 min until the cells became completely homogenous. Digestion was stopped by adding equal volume of complete medium (DMEM, containing 10% FBS). After centrifuging at 300 × g for 5 min, SVFs were in the bottom and mature adipocytes layered on the top. The oil was abandoned and the mature adipocyte layer was moved to a new 15 mL tube. After centrifuging at 100 × g for 2–3 times, the oil layer was removed and mature adipocytes were obtained. For SVFs, the cell pellet was resuspended in 5 mL medium, and passed through a 70 μm cell strainer. The cell suspension was transferred to a 15 mL tube and centrifuged at 300 × g for 5 min. SVFs were collected in the bottom of the tube.

**Flow cytometry analysis**. As above mentioned, SVFs were isolated from mouse adipose tissues. Flow cytometry was conducted as described[51]. The SVFs were then stained in cold PBS containing 0.5% BSA and 2 mM EDTA (pH 7.4) with PE anti-mouse F4/80 (123110), FITC anti-mouse CD11b (101206) and APC anti-mouse CD11c (117310) for 30 min at 4 °C. After fixed with Fixation Buffer (420801) at 4 °C for 30 min, the cells stained in Permeabilization Wash Buffer (421002) with PE/Cy7 anti-mouse CD206 (141720) at 4 °C for 30 min. All the antibodies and reagents were purchased from Biolegend, San Diego, CA, USA. The antibody dilution was 1/400. The cells were subjected to flow cytometry analysis using a MACSQuantTM (Miltenyi Biotec, Auburn, CA, USA). Data analysis was performed using the FlowJo software (Tree Star Inc, Ashland, USA) as instructed.

**Chromatin immunoprecipitation (ChIP) assay**. Adipocytes originated from 3T3-L1 cells were treated with Kyn or PBS, and then crosslinked for 10 min by 1%

formaldehyde in PBS, respectively. ChIP assays were next performed using a ChIP Assay Kit (Beyotime Biotechnology, Jiang Su, China). Polyclonal antibodies against AhR (sc-8088 X, Santa Cruz Biotechnology, CA, USA, 1/100 dilution) were employed for the assays, and a normal rabbit IgG (sc-2027 X, Santa Cruz Biotechnology, CA, USA, 1/100 dilution) was used as negative controls. Primers used for ChIP assays were as follows: Stat3- F-5'- TGT GCC TGG CTT CAA AGT A -3', and R-5'- CTG CCC ACT CCT GAT GCT G -3'.

**Luciferase reporter assays**. The promoter sequence for *Stat3* (−2873 to −1) (start codon as +1) was amplified from mouse genomic DNA. The mutated *Stat3* promoter (Stat3 Mu, the AhR binding motif TTGCGTG was mutated to CGATACA) was directly synthesized by the Tsingke Biological Technology (Beijing, China). The resulting products were sub-cloned into a pGL-3 vector (pGL3-Stat3 wt and pGL3-Stat3 mut), and then confirmed by DNA sequencing, respectively. The mutated *Il-6* promoter (Il-6 Mu, the Stat3 binding motif CTCCTGGAAA was mutated to CAGGCTCTCT) and the pGL-3 vectors (pGL3-Il-6 wt and pGL3-Il-6 mut) were synthesized by Genechem (Shanghai, China). For *Stat3* promoter reporter assays, 3T3-L1 were transfected with a mixture of plasmids for pGL3 luciferase reporter and pRL-TK luciferase (20:1). The cells were treated with 100 μg mL$^{-1}$ Kyn for 12 h following 24 h of transfection, and then harvested for analysis of luciferase activities using a dual-luciferase reporter assay system (Promega, Madison, WI, USA) according to the instruction. For *Il-6* promoter reporter assays, 3T3-L1 cells were transfected with the plasmids and treated with Kyn as above, and then stimulated with Stattic (10 μM) /StemRegenin 1 (1 μM) for 12 h.

**Statistical analysis**. Data were represented as mean ± SEM. The normality of the data was tested using the Shapiro–Wilk normality test. Data were compared using unpaired, two-tailed Student's *t* test, one-way ANOVA with the Holm–Sidak multiple-comparisons test with corrections, Spearman's correlation or two-way ANOVA followed with Bonferroni's multiple comparisons test when applicable. Statistical analyses for correlation were performed using Spearman's correlation. In all cases, $p < 0.05$ was considered as statistical significance. All statistical tests were performed by the GraphPad Prism version 5.0 software (La Jolla, CA, USA). The images were created by using PowerPoint 2016 and CorelDRAW X6 software.

**Reporting summary**. Further information on research design is available in the Nature Research Reporting Summary linked to this article.

## Data availability

All data supporting this paper are present within the paper and/or in the Supplementary Materials. Source data are provided with this paper.

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

## Acknowledgements

We are grateful to those patients for donating their blood and adipose tissues for the studies. We are also grateful to Long Cai for his help to generate a paradigm for summary of our studies, Jing Liu, Yang Li, Xi Luo, Zhu He, Wenye Mo and Siqian Liu from the Center of Biomedical Research, Tongji Hospital, Tongji Medical College, Huazhong University of Science and Technology for their help in animal studies. Our study was supported by the National Natural Science Foundation of China (82130023, 81920108009 and 91749207 to C-Y.W., 82100892 to J.Z., 82070808 to S.Z., 81873656 to F.X., 82100823 to F.W., 82100931 to H.Z., 81770823 to P.Y. and 81800068 to Y.W.), Department of Science and Technology of Hubei Province Program project (2020DCD014) to C.-Y.W., the Postdoctoral Science Foundation of China (54000-0106540081 and 54000-0106540080) to C.-Y.W., Hubei Health Committee Program (WJ2021ZH0002) to C.-Y.W., the Integrated Innovative Team for Major Human Diseases Program of Tongji Medical College, Huazhong University of Science and Technology, and the Innovative Funding for Translational Research from Tongji Hospital.

## Author contributions

T.H., J.S., J.G. and J.C. contributed equally to this work. T.H., J.S., J.G. and J.C. designed the experiments and analyzed the data. Q.Z. and T.H. wrote the manuscript. J.S. and T.H. performed the majority of the experiments. J.G., J.C., H.X., L.Z., Z.G. and X.W. were involved in RT-qPCR and western blot. J.G., H.X., Y-H.W. and L.Z. contributed toward the fractionation of adipose tissue. W.Y., J.H., S.Z. and Q.Y. assisted with the animal experiments. S.Z. contributed toward the collection of clinical samples. F.X., Q.Z., S.L. and C-Y.W. contributed to the study design and manuscript preparation.

## Competing interests

The authors declare no competing interests.
