## [Peer Review File · Nature Communications]

Adipocyte-derived kynurenine promotes obesity and insulin resistance by activating the AhR/STAT3/IL-6 signalingREVIEWER COMMENTS

Reviewer #1 (Remarks to the Author):

The paper demonstrates that IDO1 in mature adipocytes controls circulating Kyn levels, suggesting an important contribution of adipose tissue for Kyn metabolism. Adipocyte specific ablation of Ido1 reduces Kyn accumulation and protects from obesity. The authors furthermore suggest that Kyn regulates lipid homeostasis by regulating the AhR/Stat3/IL-6 axis, which in turn regulates insulin sensitivity. The study is very interesting as it identifies Kyn as an adipokine with implications in energy metabolism. There are however some points which need to be addressed to substantiate the claims made here.

Major:

As one of the STAT3 targets, IL6 is upregulated in response to AHR activation by kynurenine, which the authors demonstrate nicely. However, the study lacks a direct mechanistic evidence supporting the statement that the effect of kynurenine is indeed mediated by IL6. In addition, authors provide no explanation how the observed increase in STAT3 activity and IL6 secretion in adipose tissue would worsen systemic metabolism. Are IL6 levels increased also in the systemic circulation, or only locally within fat depots?

Do the authors see increased AHR expression, STAT3 phosphorylation and increased IL6 levels in obese mice and mice treated with kynurenine, and conversely reduced levels of these three proteins in Ido1-aKO mice? These results would at least indirectly support the hypothesis that IL6 is the key mediator of kynurenine effects in adipose tissue.

Authors should justify the Kynurenine dosage used in the mouse in vivo study. Circulating kynurenine levels in chow fed mice reach 200 ng/ml. Therefore, administration of 20 mg/kg is a very high dose, which is expected to cause a much higher increase in circulating kynurenine levels than observed in obesity. How long after kynurenine administration was its circulating level (Fig S2a) determined?

Lines 111-112: Authors should correct their statement that “depletion of Ido1 restored the plasma Kyn and KTR levels to the basal levels”. Basal levels imply levels found in plasma of control animals, and kynurenine was depleted almost completely.

Lines 116-117, 179-180, 247-248: Authors wrongly interpret an increase in RER as enhanced energy expenditure. RER is an indicator of fuel preference and therefore the observed increase should be interpreted as a shift towards carbohydrate utilization. This would also explain the observed increase in SUVmax on FDG-PET. How do the authors explain the increase in RER in HFD fed Ido1 knockout mice in Fig S2j? These mice show an RER pattern similar to chow fed mice, which is under HFD condition unexpected.

Authors should provide a rationale why they decided to investigate the role of kynurenine specifically in mature adipocytes, and why they quantified specifically IL6 expression even though STAT3 drives expression of several genes implicated in metabolism.

Importantly, statistical analysis is not performed correctly since all data were analyzed by Student's t-test as claimed in Figure Legends. This is not acceptable for analysis of 3 or more experimental groups such as Figures 1b and c, 4k and l, 6 and e or in case of dose response experiments such as Fig. 4b, c, d and f. Next, a two-way ANOVA should be applied in experiments where effect of 2 factors is compared such as diet and cell fraction (Fig. 2c) or genotype and treatment (Fig. 4i). In addition, all data obtained from glucose/insulin tolerance tests and/or measurement of body weight over the time must be analyzed by two-way ANOVA with repeated measurements followed by appropriate multiple comparison tests.

Minor:

- in compliance with good data presentation practice all y axis in plots should be set to 0 such as body weight curves (Fig. 1f, 1j, 3b, 5b and 6f), AUC calculations (Fig. 3g, 3h, 5c, 6g and 6h) or IL-6 levels (Fig. 4l and 6e)
- knockout efficiency should be provided for all mouse models
- Please, introduce the abbreviation AHR when first mentioned in the main text (line 76).
- Authors should consider renaming the “normal” group of human subjects presented in figure 1 to “lean”.
- Authors should utilize proofreading services, as there are numerous typos and wrongly used phrases in the text.
- Fig.3 a – there is a discrepancy between scheme and figure legend in terms of which exon is flanked by LoxP sites

Reviewer #2 (Remarks to the Author):

In this manuscript the authors report on the correlation between adipocyte-derived kynurenine and obesity and insulin resistance. The authors found that overexpressed indoleamine 2, 3-dioxygenase 1 (IDO1) in mature adipocytes primarily promoted the overcycling of Kyn, which further exacerbated insulin resistance by disrupting lipid homeostasis in adipocytes by activating AhR/Stat3/IL-6 signaling. Knockout of IDO1 or AhR in adipocytes eliminated the effect of Kyn. Moreover, vitamin B6 supplementation did indeed reduce the development of obesity and insulin resistance by breaking down Kyn derived from adipocytes.

The earlier part of this manuscript consists of a well-conducted series of experiments that discover the correlation between adipocyte-derived kynurenine and obesity and insulin resistance. However, the authors' attempts to describe the mechanisms that Kyn prefers to impair the lipid homeostasis of adipocytes are less convincing. Also, controversial results/conclusions are present in the manuscript.

Specific points:

1) One of the key conclusions of this manuscript concerns Kyn prefers to impair the lipid homeostasis of adipocytes via activating AhR/Stat3/IL-6 signaling, further exacerbating insulin resistance. And the role of IL-6 signaling on obesity and insulin resistance is discussed controversially. However, the authors did not confirm the effect of Stat3 and IL-6 on insulin tolerance and obesity. Whether IL-6 can directly regulate the synthesis and decomposition of fat in adipocytes?

2) In general, IL-6 is synthesized by fibroblasts, monocytes, macrophages, T cells and endothelial cells. Could macrophages in adipose tissue be induced to secrete IL-6 by kyn?

3) The liver is the main organ for regulating blood glucose. It maintains the relative stability of blood glucose through the synthesis and decomposition of liver glycogen. What is the effect of kyn up-regulation in peripheral blood on liver glycogen?

4) Compared with SR-1, the phosphorylation of Stat3 was significantly inhibited by Stattic, but the inhibitory effect on IL-6 secretion was similar to that of SR-1(Fig. 4k, l). Please explain the reason.

5) The authors indicate that high levels of circulating Kyn in overweight/obese subjects are likely caused by the enhanced IDO1 expression. What are the main factors up-regulating IDO1 in HFD-fed or obesity? In HFD-fed, IFN- γ , known as a potent inducer of IDO, was up-regulated in plasma or adipose tissue (Reference 9). Does IFN- γ play a role in the AhR/STAT3/IL-6 signaling?

6) Although the article provided data on KYN down-regulating lipolysis (up-regulating p-HSL) and promoting lipogenesis (down-regulating P-ACC), it did not provide direct data on whether AHR affected fat metabolism.

7) The weight of PBS group was about 32g at day 0 (8 weeks of high-fat diet) (Fig. 1f); The weight of WT group was about 42g at 8 weeks of high-fat diet (Fig. 1j). The two groups were WT mice on a high-fat

diet for 8 weeks with a weight difference of about 10g. What is the reason for the large difference between batches?

8) In Fig. 1j-m shows the difference of weight, GTT and ITT between WT group and Ido1 - / - group under high-fat diet. The difference of weight, GTT and ITT between WT group and Ido1 - / -group under normal diet should be added to exclude the influence of other factors.

9) The decrease in Stat3 in adipocytes from Ahr-aKO mice is not clear, however, the decrease in p-Stat3 were significantly in adipocytes from Ahr-aKO mice (Fig. 5i). And the disruption of binding motifs (mutant plasmids) leads to inadequate transcription reduction (Fig. 4i). Please explain the reason. How is Stat3 phosphorylated in adipocytes?

Reviewer #3 (Remarks to the Author):

The manuscript entitled "Adipocyte-Derived Kynurenine Promotes Obesity and Insulin Resistance by Activating the AhR/STAT3/IL-6 Signaling" addressed the role of kynurenine in adipocyte function, insulin resistance, and obesity. The authors have combined several genetic and pharmacological approaches to show that adipocyte-derived Kyn directly regulates IL6 expression through AhR and STAT3 expression. This is a novel finding and is highly relevant to the field of adipose biology. However, the findings of the paper are contradictory to several previous studies about the functions of the Kynurenine pathway in adipose tissue. The manuscript would benefit from discussing those studies (including Agudelo et al, Cell metabolism 2018) and explaining the potential reasons for the contradictory findings.

In studies presented in figure 1, how was the Kyn dose chosen?

Since the concentration of Kyn in the liver and skeletal muscle is higher than WAT, it is surprising the loss of the adipocyte Ido has such a major impact on plasma Kyn. Can the authors explain this?

The measurement of heat production and energy expenditure should not be normalized to the body weight. Particularly in this study, the difference in the body weight between the groups complicates the interpretation of the data if normalized to the body weight. The regression analysis of the energy expenditure data provides a more accurate understanding of the mass-dependent and independent

effects on the energy expenditure parameters and should be included (Timo D. Mueller, Nature Metabolism 2021).

In referring to figure 3e, the authors note that the increase in respiratory exchange ratio (RER) represents changes in the energy expenditure. RER is the ratio between the metabolic production of carbon dioxide (CO₂) and the uptake of oxygen (O₂) and provides information about fuel choice and utilization.

Related to figure 3e and figure 5f, what is the reason for the increase in RER in Ido1-aKO and Ahr-aKO mice?

On page 9, “in silicon” should be replaced with “in silico”.

The STAT3 blot in figure 4k seems to have been modified. The authors should provide the original western blot images for all the figure panels.

The words “surprisingly, interestingly, impressively, etc” are over-used in the manuscript and should be removed.

Reviewers' Comments:

Reviewer #1 (Remarks to the Author):

The paper demonstrates that IDO1 in mature adipocytes controls circulating Kyn levels, suggesting an important contribution of adipose tissue for Kyn metabolism. Adipocyte specific ablation of Ido1 reduces Kyn accumulation and protects from obesity. The authors furthermore suggest that Kyn regulates lipid homeostasis by regulating the AhR/Stat3/IL-6 axis, which in turn regulates insulin sensitivity. The study is very interesting as it identifies Kyn as an adipokine with implications in energy metabolism. There are however some points which need to be addressed to substantiate the claims made here.

Major:

As one of the STAT3 targets, IL6 is upregulated in response to AHR activation by kynurenine, which the authors demonstrate nicely. However, the study lacks a direct mechanistic evidence supporting the statement that the effect of kynurenine is indeed mediated by IL6. In addition, authors provide no explanation how the observed increase in STAT3 activity and IL6 secretion in adipose tissue would worsen systemic metabolism. Are IL6 levels increased also in the systemic circulation, or only locally within fat depots?

Do the authors see increased AHR expression, STAT3 phosphorylation and increased IL6 levels in obese mice and mice treated with kynurenine, and conversely reduced levels of these three proteins in Ido1-aKO mice? These results would at least indirectly support the hypothesis that IL6 is the key mediator of kynurenine effects in adipose tissue.

Authors should justify the Kynurenine dosage used in the mouse in vivo study. Circulating kynurenine levels in chow fed mice reach 200 ng/ml. Therefore, administration of 20 mg/kg is a very high dose, which is expected to cause a much higher increase in circulating kynurenine levels than observed in obesity. How long after kynurenine administration was its circulating level (Fig S2a) determined? Lines 111-112: Authors should correct their statement that "depletion of Ido1 restored the plasma Kyn and KTR levels to the basal levels". Basal levels imply levels found in plasma of control animals, and kynurenine was depleted almost completely.

Lines 116-117, 179-180, 247-248: Authors wrongly interpret an increase in RER as enhanced energy expenditure. RER is an indicator of fuel preference and therefore the observed increase should be interpreted as a shift towards carbohydrate utilization. This would also explain the observed increase in SUVmax on FDG-PET. How do the authors explain the increase in RER in HFD fed Ido1 knockout mice in Fig S2j? These mice show an RER pattern similar to chow fed mice, which is under HFD condition unexpected.

Authors should provide a rationale why they decided to investigate the role of kynurenine specifically in mature adipocytes, and why they quantified specifically IL6 expression even though STAT3 drives expression of several genes implicated in metabolism.

Importantly, statistical analysis is not performed correctly since all data were analyzed by Student's t-test as claimed in Figure Legends. This is not acceptable for analysis of 3 or more experimental groups such as Figures 1b and c, 4k and l, 6 and e or in case of dose response experiments such as Fig. 4b, c, d and f. Next, a two-way ANOVA should be applied in experiments where effect of 2 factors is compared such as diet and cell fraction (Fig. 2c) or genotype and treatment (Fig. 4i). In addition, all data obtained from glucose/insulin tolerance tests and/or measurement of body weight over the time must be analyzed by two-way ANOVA with repeated measurements followed by appropriate multiple comparison tests.

Minor:

- in compliance with good data presentation practice all y axis in plots should be set to 0 such as body weight curves (Fig. 1f, 1j, 3b, 5b and 6f), AUC calculations (Fig. 3g, 3h, 5c, 6g and 6h) or IL-6 levels (Fig. 4l and 6e)
- knockout efficiency should be provided for all mouse models
- Please, introduce the abbreviation AHR when first mentioned in the main text (line 76).
- Authors should consider renaming the “normal” group of human subjects presented in figure 1 to “lean”.
- Authors should utilize proofreading services, as there are numerous typos and wrongly used phrases in the text.
- Fig.3 a – there is a discrepancy between scheme and figure legend in terms of which exon is flanked by LoxP sites

Reviewer #2 (Remarks to the Author):

In this manuscript the authors report on the correlation between adipocyte-derived kynurenine and obesity and insulin resistance. The authors found that overexpressed indoleamine 2, 3-dioxygenase 1 (IDO1) in mature adipocytes primarily promoted the overcycling of Kyn, which further exacerbated insulin resistance by disrupting lipid homeostasis in adipocytes by activating AhR/Stat3/IL-6 signaling. Knockout of IDO1 or AhR in adipocytes eliminated the effect of Kyn. Moreover, vitamin B6 supplementation did indeed reduce the development of obesity and insulin resistance by breaking down Kyn derived from adipocytes.

The earlier part of this manuscript consists of a well-conducted series of experiments that discover the correlation between adipocyte-derived kynurenine and obesity and insulin resistance. However, the authors' attempts to describe the mechanisms that Kyn prefers to impair the lipid homeostasis of adipocytes are less convincing. Also, controversial results/conclusions are present in the manuscript.

Specific points:

- 1) One of the key conclusions of this manuscript concerns Kyn prefers to impair the lipid homeostasis of adipocytes via activating AhR/Stat3/IL-6 signaling, further exacerbating insulin resistance. And the role of IL-6 signaling on obesity and insulin resistance is discussed controversially. However, the authors did not confirm the effect of Stat3 and IL-6 on insulin tolerance and obesity. Whether IL-6 can directly regulate the synthesis and decomposition of fat in adipocytes?
- 2) In general, IL-6 is synthesized by fibroblasts, monocytes, macrophages, T cells and endothelial cells. Could macrophages in adipose tissue be induced to secrete IL-6 by kyn?
- 3) The liver is the main organ for regulating blood glucose. It maintains the relative stability of blood glucose through the synthesis and decomposition of liver glycogen. What is the effect of kyn up-regulation in peripheral blood on liver glycogen?
- 4) Compared with SR-1, the phosphorylation of Stat3 was significantly inhibited by Stattic, but the inhibitory effect on IL-6 secretion was similar to that of SR-1(Fig. 4k, l). Please explain the reason.
- 5) The authors indicate that high levels of circulating Kyn in overweight/obese subjects are likely caused by the enhanced IDO1 expression. What are the main factors up-regulating IDO1 in HFD-fed or obesity? In HFD-fed, IFN- γ , known as a potent inducer of IDO, was up-regulated in plasma or adipose tissue (Reference 9). Does IFN- γ play a role In the AhR/STAT3/IL-6 signaling?
- 6) Although the article provided data on KYN down-regulating lipolysis (up -regulating p-HSL) and promoting lipogenesis (down-regulating P-ACC), it did not provide direct data on whether AHR affected fat metabolism.
- 7) The weight of PBS group was about 32g at day 0 (8 weeks of high-fat diet) (Fig. 1f); The weight of

WT group was about 42g at 8 weeks of high-fat diet (Fig. 1j). The two groups were WT mice on a high-fat diet for 8 weeks with a weight difference of about 10g. What is the reason for the large difference between batches?

8) In Fig. 1j-m shows the difference of weight, GTT and ITT between WT group and Ido1 - / - group under high-fat diet. The difference of weight, GTT and ITT between WT group and Ido1 - / -group under normal diet should be added to exclude the influence of other factors.

9) The decrease in Stat3 in adipocytes from Ahr-aKO mice is not clear, however, the decrease in p-Stat3 were significantly in adipocytes from Ahr-aKO mice (Fig. 5i). And the disruption of binding motifs (mutant plasmids) leads to inadequate transcription reduction (Fig. 4i). Please explain the reason. How is Stat3 phosphorylated in adipocytes?

Reviewer #3 (Remarks to the Author):

The manuscript entitled "Adipocyte-Derived Kynurenine Promotes Obesity and Insulin Resistance by Activating the AhR/STAT3/IL-6 Signaling" addressed the role of kynurenine in adipocyte function, insulin resistance, and obesity. The authors have combined several genetic and pharmacological approaches to show that adipocyte-derived Kyn directly regulates IL6 expression through AhR and STAT3 expression. This is a novel finding and is highly relevant to the field of adipose biology. However, the findings of the paper are contradictory to several previous studies about the functions of the Kynurenine pathway in adipose tissue. The manuscript would benefit from discussing those studies (including Agudelo et al, Cell metabolism 2018) and explaining the potential reasons for the contradictory findings.

In studies presented in figure 1, how was the Kyn dose chosen?

Since the concentration of Kyn in the liver and skeletal muscle is higher than WAT, it is surprising the loss of the adipocyte Ido has such a major impact on plasma Kyn. Can the authors explain this?

The measurement of heat production and energy expenditure should not be normalized to the body weight. Particularly in this study, the difference in the body weight between the groups complicates the interpretation of the data if normalized to the body weight. The regression analysis of the energy expenditure data provides a more accurate understanding of the mass-dependent and independent effects on the energy expenditure parameters and should be included (Timo D. Mueller, Nature Metabolism 2021).

In referring to figure 3e, the authors note that the increase in respiratory exchange ratio (RER) represents changes in the energy expenditure. RER is the ratio between the metabolic production of carbon dioxide (CO₂) and the uptake of oxygen (O₂) and provides information about fuel choice and utilization.

Related to figure 3e and figure 5f, what is the reason for the increase in RER in Ido1-aKO and Ahr-aKO mice?

On page 9, "in silicon" should be replaced with "in silico".

The STAT3 blot in figure 4k seems to have been modified. The authors should provide the original western blot images for all the figure panels.

The words "surprisingly, interestingly, impressively, etc" are over-used in the manuscript and should be removed.

Point-by-point responses to reviewers:

Reviewer 1:

The paper demonstrates that IDO1 in mature adipocytes controls circulating Kyn levels, suggesting an important contribution of adipose tissue for Kyn metabolism. Adipocyte specific ablation of *Ido1* reduces Kyn accumulation and protects from obesity. The authors furthermore suggest that Kyn regulates lipid homeostasis by regulating the AhR/Stat3/IL-6 axis, which in turn regulates insulin sensitivity. The study is very interesting as it identifies Kyn as an adipokine with implications in energy metabolism. There are however some points which need to be addressed to substantiate the claims made here.

Major:

1. As one of the STAT3 targets, IL6 is upregulated in response to AHR activation by kynurenine, which the authors demonstrate nicely. However, the study lacks a direct mechanistic evidence supporting the statement that the effect of kynurenine is indeed mediated by IL6.

Responses: This question is highly appreciated, which would improve our manuscript a lot. In order to address this question, both *in vitro* and *in vivo* rescue experiments were conducted to further corroborate that the effect of Kyn is mediated by IL-6. In the rescue experiments, Tocilizumab (TCZ), an IL-6 receptor blocking antibody, was applied to block the effect of IL-6. Indeed, we discovered that TCZ could attenuate the impact caused by Kyn, indicating that the effect of Kyn is mediated by IL-6. The data are shown in Fig. 5 and related descriptions are added to the manuscript.

2. In addition, authors provide no explanation how the observed increase in STAT3 activity and IL6 secretion in adipose tissue would worsen systemic metabolism. Are IL6 levels increased also in the systemic circulation, or only locally within fat depots?

Responses: This is a constructive suggestion and, we thus detected the plasma IL-6 levels of the mice. As shown in Fig. S7f, compared to that of normal diet fed mice, the plasma IL-6 level was increased in high fat diet fed mice, which was even higher in HFD+Kyn group. Similar tendency was also observed in the WAT IL-6 levels (Fig. S7g). To verify whether the systemic metabolism was affected, we detected insulin sensitivities in the liver and skeletal muscle, as well as the synthesis of liver glycogen. The results (Fig. S1h and i) indicated that Kyn administration further exacerbated insulin resistance in the liver and muscle, coupled with an enhanced gluconeogenesis and attenuated synthesis of liver glycogen.

3. Do the authors see increased AHR expression, STAT3 phosphorylation and increased IL6 levels in obese mice and mice treated with kynurenine, and conversely reduced levels of these three proteins in *Ido1*-aKO mice? These results would at least indirectly support the hypothesis that IL6 is the key mediator of kynurenine effects in adipose tissue.

Responses: Thanks for your suggestion. To answer this question, we evaluated the expression of AhR and phosphorylation of STAT3, as well as the plasma and WAT IL-6 levels. As shown in Fig. S7e-g, Kyn would activate the AhR/STAT3/IL-6 axis, while this effect was abolished in *Ido1*-aKO mice (Fig. S7a-d). Combined with the above rescue experiments (Fig. 5), these results strongly substantiated the hypothesis that IL6 is the key mediator of kynurenine effects in adipose tissue, which improve the manuscript a lot.

4. Authors should justify the Kynurenine dosage used in the mouse in vivo study. Circulating kynurenine levels in chow fed mice reach 200 ng/ml. Therefore, administration of 20 mg/kg is a very high dose, which is expected to cause a much higher increase in circulating kynurenine levels than observed in obesity. How long after kynurenine administration was its circulating level (Fig S2a) determined?

Responses: At the very beginning, we considered the increase of Kyn in obese subjects as a compensatory mechanism. We thus employed Kyn as an anti-obesity treatment and injected mice with a high dosage of Kyn to expect a therapeutic effect. Before the injection, we had done a pharmacokinetic study and now added the data to Fig. S1d. As shown in Fig. S1d, the plasma concentration of Kyn reached to 800 ng/ml following 1h of injection, and then gradually decreased to around 370 ng/ml after 24h of injection, which just reached to the circulating Kyn concentration detected in obese mice (Fig. 1d, 400 ng/ml). After administrating the mice with Kyn and HFD for one month, the plasma concentration of Kyn was around 500 ng/ml, only a little bit higher than that in HFD fed mice (Fig. S1e). Therefore, this dosage is not very high for the mice.

5. Lines 111-112: Authors should correct their statement that “depletion of Ido1 restored the plasma Kyn and KTR levels to the basal levels”. Basal levels imply levels found in plasma of control animals, and kynurenine was depleted almost completely.

Responses: We are sorry for the mistake. We have corrected the statement in the revised version.

6. Lines 116-117, 179-180, 247-248: Authors wrongly interpret an increase in RER as enhanced energy expenditure. RER is an indicator of fuel preference and therefore the observed increase should be interpreted as a shift towards carbohydrate utilization. This would also explain the observed increase in SUVmax on FDG-PET. How do the authors explain the increase in RER in HFD fed Ido1 knockout mice in Fig S2j? These mice show an RER pattern similar to chow fed mice, which is under HFD condition unexpected.

Responses: Many thanks for pointing out these issues. We have corrected the interpretation for the RER data in the revised manuscript. Based on the suggestion, we added normal chow fed WT mice as a negative control and measure the RER of these mice. The renewed data was updated in Fig. S2k. Indeed, the RER of HFD fed *Ido1* knockout mice was higher than that of HFD fed WT mice, which indicated a shift of energy source towards carbohydrate utilization. Meanwhile, the RER of HFD fed *Ido1* knockout mice was lower than that of chow fed WT mice (Fig. S2j), just as the reviewer pointed out.

7. Authors should provide a rationale why they decided to investigate the role of kynurenine specifically in mature adipocytes, and why they quantified specifically IL6 expression even though STAT3 drives expression of several genes implicated in metabolism.

Responses: We regret for not highlighting enough for the rationale why we decided to investigate the role of kynurenine specifically in mature adipocytes. We added the explanation in the revised

manuscript.

For IL-6, we added Fig. S7a-c to explain why we chose IL-6 as a target gene. The above results indicated that the effect of adipocyte-specific *Ido1* knockout also impaired the systemic metabolism, which prompted us to assume that this effect could be caused by the excessive production of circulating cytokines. We thus first conducted RT-PCR to assess the expression differences of inflammatory cytokines between HFD fed *Ido1*-aKO mice and WT mice (Fig. S7a). IL-6 was identified as the most significant one, which was further confirmed by ELISA analyses of plasma and WAT samples (Fig. S7b and c). Therefore, IL-6 was selected as the downstream target gene of STAT3. We have added the above information into the revised manuscript.

8. Importantly, statistical analysis is not performed correctly since all data were analyzed by Student's t-test as claimed in Figure Legends. This is not acceptable for analysis of 3 or more experimental groups such as Figures 1b and c, 4k and l, 6 and e or in case of dose response experiments such as Fig. 4b, c, d and f. Next, a two-way ANOVA should be applied in experiments where effect of 2 factors is compared such as diet and cell fraction (Fig. 2c) or genotype and treatment (Fig. 4i). In addition, all data obtained from glucose/insulin tolerance tests and/or measurement of body weight over the time must be analyzed by two-way ANOVA with repeated measurements followed by appropriate multiple comparison tests.

Responses: Thanks for pointing out this defect. All of our data in the revised manuscript were reanalyzed by a professional statistician. Detailed approaches for data analysis were described in the method part.

Minor:

1. In compliance with good data presentation practice all y axis in plots should be set to 0 such as body weight curves (Fig. 1f, 1j, 3b, 5b and 6f), AUC calculations (Fig. 3g, 3h, 5c, 6g and 6h) or IL-6 levels (Fig. 4l and 6e).

Responses: We really appreciate this friendly reminder, and have revised the figures according to the suggestion.

2. Knockout efficiency should be provided for all mouse models.

Responses: As requested, we have provided the RT-PCR results of *Ido1*^{-/-}, *Ido1*-IKO and *Ido1*-aKO mice to assess the knockout efficiency of *Ido1* (Fig. S10a-c). The primers employed were listed in Table S3. The knockout efficiency of AhR in *Ahr*-aKO mice was tested by western blot, and the result was shown in Fig. 6i. All above results indicated a sufficient knockout of the target genes.

3. Please, introduce the abbreviation AHR when first mentioned in the main text (line 76)

Responses: We added the full name of AhR when it was first mentioned in the main text.

4. Authors should consider renaming the “normal” group of human subjects presented in figure 1 to “lean”.

Responses: Thanks for the suggestion. We have changed the text from “normal” to “lean” in Figure 1, Figure 2, Figure 7a, Figure S1 and Table S1, as well as in the main text.

5. Authors should utilize proofreading services, as there are numerous typos and wrongly used phrases in the text.

Responses: As requested, we utilized a proofreading service for the revised manuscript and corrected the typos and wrongly used phrases.

6. Fig.3 a – there is a discrepancy between scheme and figure legend in terms of which exon is flanked by LoxP sites.

Responses: We are sorry for this mistake. We corrected the description in the figure legend.

Reviewer 2:

In this manuscript the authors report on the correlation between adipocyte-derived kynurenine and obesity and insulin resistance. The authors found that overexpressed indoleamine 2, 3-dioxygenase 1 (IDO1) in mature adipocytes primarily promoted the overcycling of Kyn, which further exacerbated insulin resistance by disrupting lipid homeostasis in adipocytes by activating AhR/Stat3/IL-6 signaling. Knockout of IDO1 or AhR in adipocytes eliminated the effect of Kyn. Moreover, vitamin B6 supplementation did indeed reduce the development of obesity and insulin resistance by breaking down Kyn derived from adipocytes.

The earlier part of this manuscript consists of a well-conducted series of experiments that discover the correlation between adipocyte-derived kynurenine and obesity and insulin resistance. However, the authors' attempts to describe the mechanisms that Kyn prefers to impair the lipid homeostasis of adipocytes are less convincing. Also, controversial results/conclusions are present in the manuscript.

Specific points:

1. One of the key conclusions of this manuscript concerns Kyn prefers to impair the lipid homeostasis of adipocytes via activating AhR/Stat3/IL-6 signaling, further exacerbating insulin resistance. And the role of IL-6 signaling on obesity and insulin resistance is discussed controversially. However, the authors did not confirm the effect of Stat3 and IL-6 on insulin tolerance and obesity. Whether IL-6 can directly regulate the synthesis and decomposition of fat in adipocytes?

Responses: We really appreciate this constructive suggestion, and this is a similar question raised by Reviewer 1. To further verify the effect of Kyn-induced IL-6 in regulating the synthesis and decomposition of fat in adipocytes, we conducted both *in vitro* and *in vivo* rescue experiments. Due to the source of IL-6 might affect the physiological metabolic response^{1,2}, we preferred not to use the exogenous IL-6 to stimulate the adipocyte. Instead, Tocilizumab (TCZ), an IL-6 receptor blocking antibody, was applied to block the effect of adipocyte-derived endogenous IL-6 in the rescue experiments. Indeed, we discovered that the treatment of TCZ could ameliorate the impact caused by Kyn-induced IL-6, indicating that the effect of Kyn is mediated by IL-6. The data are shown in Fig. 5 and the related descriptions are added to the manuscript. As addressed earlier, we also examined the expression of AhR and phosphorylation of STAT3, as well as the plasma and WAT IL-6 levels. As shown in Fig. S7e-g, Kyn would activate the AhR/STAT3/IL-6 axis, while this effect was abolished in *Ido1*-aKO mice (Fig. S7a-d).

Indeed, the role of IL-6 signaling in obesity and insulin resistance is yet to be fully elucidated, even controversial results had been reported. However, recent studies have demonstrated that the controversial effect of IL-6 signaling on obesity and insulin resistance might be attributed to its source¹. As reported, IL-6 secreted by adipocytes during obesity would promote chronic inflammation and exacerbate metabolic syndrome^{1,2}, which is consistent with our studies. Relevant discussion has been added into the manuscript, which really improves our manuscript a lot. Thanks again.

1. Han MS, White A, Perry RJ, et al. Regulation of adipose tissue inflammation by interleukin 6. *Proceedings of the National Academy of Sciences*, 2020, 117(6):201920004.

2. Kistner TM, Pedersen BK, Lieberman DE. Interleukin 6 as an energy allocator in muscle tissue. *Nature Metabolism*, 2022.

2. In general, IL-6 is synthesized by fibroblasts, monocytes, macrophages, T cells and endothelial cells. Could macrophages in adipose tissue be induced to secrete IL-6 by kyn?

Responses: This is a very critical question. In general, IL-6 is produced by M1 macrophages. To address this question, we firstly induced M1 macrophages using bone-marrow derived macrophages (BMDM). The polarized M1 macrophages were stimulated with different concentrations of Kyn. Unexpectedly, at the concentration of 50 µg/ml and 100 µg/ml Kyn, the production of IL-6 by M1 macrophages tended to be reduced (Fig. S7i-j), which was opposite to what we observed in adipocytes. This interesting phenomenon was also observed by Wang and colleagues, in which the production of IL-6 was blunted in LPS-induced THP-1 following Kyn stimulation³. Although Kyn attenuated macrophage-derived IL-6, it did not affect our previous conclusion that Kyn would exacerbate the metabolic syndrome. As reported by Han et al.¹, macrophage-derived IL-6 play a positive role in attenuating the chronic inflammation, while adipocyte-derived IL-6 just worked on an opposite way, which further supported our data. Relevant discussion has been added into the manuscript.

3. Wang D, Li D, Zhang Y, et al. Functional metabolomics reveal the role of AHR/GPR35 mediated kynurenic acid gradient sensing in chemotherapy-induced intestinal damage. *Acta Pharmaceutica Sinica B*, 2020.

3. The liver is the main organ for regulating blood glucose. It maintains the relative stability of blood glucose through the synthesis and decomposition of liver glycogen. What is the effect of kyn up-regulation in peripheral blood on liver glycogen?

Responses: To answer this question, we compared the key enzymes for liver glycogen between HFD fed mice and HFD+Kyn treated mice. As shown in Fig. S1i, increased G6pase and PEPCK along with decreased p-GSK3 β ^{Ser9} were observed in HFD+Kyn treated mice, indicating that upregulation of circulating Kyn enhances gluconeogenesis and attenuates glycogen synthesis in the liver, thereby leading to higher blood glucose levels during the course of obesity.

4. Compared with SR-1, the phosphorylation of Stat3 was significantly inhibited by Stattic, but the inhibitory effect on IL-6 secretion was similar to that of SR-1 (Fig. 4k, l). Please explain the reason.

Responses: Many thanks for pointing out this issue and we apologize for this carelessness. We, therefore, have repeated the experiments and compared the phosphorylation of STAT3 between the SR-1 treated group and Stattic treated group. As shown in the renewed data (Fig. 4j), the decrease of p-STAT3 in SR-1 treated group was just comparative to that in Stattic treated group, which can explain the inhibitory effect on IL-6 secretion that was similar in these two groups. To further corroborate this result, we conducted luciferase reporter assays. SR1 exhibited a considerable effect on inhibiting the expression of IL-6 as Stattic (Fig. S7h).

5. The authors indicate that high levels of circulating Kyn in overweight/obese subjects are likely caused by the enhanced IDO1 expression. What are the main factors up-regulating IDO1 in HFD-fed or obesity? In HFD-fed, IFN- γ , known as a potent inducer of IDO, was up-regulated in plasma or adipose tissue (Reference 9). Does IFN- γ play a role in the AhR/STAT3/IL-6 signaling?

Responses: According to the suggestion, we first examined IFN- γ levels in the WAT by ELISA. As expected, a significant increase of IFN- γ was observed in the WAT of HFD fed mice (Fig. S7k). *In vitro* studies were next conducted to address the role of IFN- γ in the AhR/STAT3/IL-6 signaling. As shown in Fig. S7l-n, IFN- γ induced IDO1 overexpression in adipocytes and, subsequently activated the AhR/STAT3/IL-6 signaling. Those results confirmed the assumption proposed by the reviewer, which is highly appreciated.

6. Although the article provided data on KYN down-regulating lipolysis (up -regulating p-HSL) and promoting lipogenesis (down-regulating P-ACC), it did not provide direct data on whether AHR affected fat metabolism.

Responses: As requested, we tested the p-HSL and p-ACC in the WAT of *AhR*-aKO mice. The results indicated that *AhR* deficiency suppressed lipogenesis and enhanced lipolysis in the WAT (Fig. S8b).

7. The weight of PBS group was about 32g at day 0 (8 weeks of high-fat diet) (Fig. 1f); The weight of WT group was about 42g at 8 weeks of high-fat diet (Fig. 1j). The two groups were WT mice on a high-fat diet for 8 weeks with a weight difference of about 10g. What is the reason for the large difference between batches?

Responses: We apologize for this mistake. The mice in Figure 1f were fed during the period of COVID-19 pandemic (01-2020 to 04-2020), by then we were quarantined at home. Therefore, we cannot ensure the timely and regular feeding of mice at that time. This batch of mice (Fig. 1f) was not as good as that of mice cared in Fig. 1j, but a tendency could still be observed in Fig. 1f. To correct this mistake, we re-conducted the experiments for both groups of mice treated with PBS and Kyn, respectively. The resulting data were employed to replace previous data presented in Fig. 1f-i and Fig. S1d-i. Those renewed figures with repeated data are much more evident indicating a deteriorated effect of Kyn on metabolic syndrome.

8. In Fig. 1j-m shows the difference of weight, GTT and ITT between WT group and *Ido1* - / - group under high-fat diet. The difference of weight, GTT and ITT between WT group and *Ido1* - / -group under normal diet should be added to exclude the influence of other factors.

Responses: The data derived from ND fed *Ido1*^{-/-} mice and WT mice were added into the figures according to the suggestion. There was no significant difference in terms of body weight between these two groups of mice, and similarly for GTT, ITT and RER tests (Fig. S2g-j).

9. The decrease in Stat3 in adipocytes from *Ahr*-aKO mice is not clear, however, the decrease in p-Stat3 were significantly in adipocytes from *Ahr*-aKO mice (Fig. 5i). And the disruption of binding motifs (mutant plasmids) leads to inadequate transcription reduction (Fig. 4i). Please explain the reason. How is Stat3 phosphorylated in adipocytes?

Responses: We appreciate very much for pointing out this defect. In Fig. 5i, the unsatisfactory for the decreased STAT3 in *Ahr*-aKO mice-derived adipocytes was likely attributed to the low purity of adipocytes. The samples could be mixed with other types of cells such as adipose tissue macrophages, which also expressed STAT3. To isolate purer adipocytes, we optimized the method according to the previous research⁴. New data resulted from repeated experiments were updated into Fig. 6i, which was consistent with *in vitro* studies.

For the luciferase report assays in Fig. 4i, we carefully checked the raw data and found out that the cells without any treatment were mistakenly employed as the mock group during data analysis. To ensure the accuracy of all data, we re-conducted all of these experiments in cells transfected with pGL3-Stat3 WT and pGL3-Stat3 MU and then stimulated with Kyn, and those transfected cells without Kyn stimulation were served as the mock group. Those new data were updated into Fig. 4h, which exhibited comparable luciferase activities between the pGL3-Stat3 MU+Kyn group and mock groups (pGL3-Stat3 WT and pGL3-Stat3 MU).

4. Zhao GN, Tian ZW, Tian T, et al. TMBIM1 is an inhibitor of adipogenesis and its depletion promotes adipocyte hyperplasia and improves obesity-related metabolic disease. *Cell Metabolism*, 2021.

Reviewer 3:

1. The manuscript entitled " Adipocyte-Derived Kynurenine Promotes Obesity and Insulin Resistance by Activating the AhR/STAT3/IL-6 Signaling" addressed the role of kynurenine in adipocyte function, insulin resistance, and obesity. The authors have combined several genetic and pharmacological approaches to show that adipocyte-derived Kyn directly regulates IL6 expression through AhR and STAT3 expression. This is a novel finding and is highly relevant to the field of adipose biology. However, the findings of the paper are contradictory to several previous studies about the functions of the Kynurenine pathway in adipose tissue. The manuscript would benefit from discussing those studies (including Agudelo et al, Cell metabolism 2018) and explaining the potential reasons for the contradictory findings.

Responses: Many thanks for your opinions. A previous study in limited number of patients suggested an increase of plasma Kyn in obese patients⁵. It is worthy of note that our current studies employed a larger cohort confirmed a significant increase of Kyn in obese patients and mice (Fig. 1a-d). More importantly, we also demonstrated convincing evidence that obesity is coupled with the deficiency of PLP (activated form of vitamin B6), a key coenzyme to catalyze Kyn to Kyna⁶, both in humans and mice (Fig. 7a-c). Indeed, Agudelo et al. reported that exercised skeletal muscle can increase the catabolism of Kyn to Kyna⁷, thereby increasing energy expenditure by activating Gpr35 and further improving energy metabolism in mice fed with high-fat diet⁸. Their data in fact are consistent with our results, in which alleviation of Kyn accumulation would improve metabolic homeostasis. Additional studies also addressed the role of Kyn in obese pathogenesis^{9,10}. Based on the suggestion, we have discussed those issues in the revised manuscript.

5. Wolowczuk I, Hennart B, Leloire A, et al. Tryptophan metabolism activation by indoleamine 2,3-dioxygenase in adipose tissue of obese women: an attempt to maintain immune homeostasis and vascular tone. *American Journal of Physiology Regulatory Integrative & Comparative Physiology*, 2012, 303(2):R135.

6. Ping S, Ramprasath T, Wang H, et al. Abnormal kynurenine pathway of tryptophan catabolism in cardiovascular diseases. *Cellular and Molecular Life Sciences*, 2017, 74(16):1-18.

7. Agudelo L, Femenía, Teresa, Orhan F, et al. Skeletal muscle PGC-1 α 1 modulates kynurenine metabolism and mediates resilience to stress-induced depression. *Cell*, 2014, 159(1):33-45.

8. Agudelo L, Femenía, Cervenka I, et al. Kynurenic acid and gpr35 regulate adipose tissue energy homeostasis and inflammation. *Cell Metabolism*, 2018, 27(2):378-392.

9. Ludivine L, Nicolas V, Yacine H, et al. Genetic deficiency of Indoleamine 2,3-dioxygenase promotes gut microbiota-mediated metabolic health. *Nature Medicine*, 2018.

10. Rojas IY, Moyer BJ, Ringelberg CS, et al. Kynurenine - induced aryl hydrocarbon receptor signaling in mice causes body mass gain, liver steatosis, and hyperglycemia. *Obesity*, 2021, 29(2).

2. In studies presented in figure 1, how was the Kyn dose chosen?

Responses: This is a similar question raised by reviewer 1 (Question 4), and we have extensively addressed it earlier.

3. Since the concentration of Kyn in the liver and skeletal muscle is higher than WAT, it is surprising the loss of the adipocyte Ido has such a major impact on plasma Kyn. Can the authors explain this?

Responses: Thanks for pointing out this issue. Indeed, the concentrations of Kyn in the liver and skeletal muscle were higher than that in the WAT (Fig. 2a). However, unlike the observations noted in the plasma and WAT, Kyn concentrations in the liver and skeletal muscle did not show a significant difference between HFD and NCD fed mice (Fig. 2a), which might be attributed to the expression differences of isozymes for Trp catabolism in these organs¹¹. In liver and skeletal muscle, TDO is predominantly expressed, which is in charge for maintaining the basal level of Kyn. Compared to TDO, IDO1 is inducible and more sensitive to the pathological insults, particularly in the setting of obesity^{6,10}. For example, during the course of HFD challenge, IDO1 was dramatically overexpressed in the WAT (Fig. 2b) accompanied by increased accumulation of Kyn (Fig. 2a), while no perceptible difference was noted in the liver and skeletal muscle between HFD and NCD fed mice (Fig. 2a and b). The increase of Kyn in WAT during obesity could also be logically explained by the discrepancy noted in terms of PLP levels (Fig. 7c), a key coenzyme to catalyze Kyn catabolism. In sharp contrast, the PLP level in the liver or skeletal muscle even exhibited a compensatory elevation in obesity (Fig. S9a-b). Collectively, the overexpressed IDO1 along with the decreased PLP synergistically leads to an elevation of Kyn in WAT during obesity, subsequently boosting the plasma Kyn. We have discussed this issue in the revised manuscript.

11. Dounay AB, Tuttle JB, Verhoest PR. Challenges and opportunities in the discovery of new therapeutics targeting the kynurenine pathway. *Journal of Medicinal Chemistry*, 2015:8762-82.

4. The measurement of heat production and energy expenditure should not be normalized to the body weight. Particularly in this study, the difference in the body weight between the groups complicates the interpretation of the data if normalized to the body weight. The regression analysis of the energy expenditure data provides a more accurate understanding of the mass-dependent and independent effects on the energy expenditure parameters and should be included (Timo D. Mueller, Nature Metabolism 2021).

Responses: We really appreciate this constructive suggestion. We have reanalyzed the energy expenditure data as suggested. The data were updated into Fig. 3e and Fig. 6f.

5. In referring to figure 3e, the authors note that the increase in respiratory exchange ratio (RER) represents changes in the energy expenditure. RER is the ratio between the metabolic production of carbon dioxide (CO₂) and the uptake of oxygen (O₂) and provides information about fuel choice and utilization.

Responses: Thanks for your explanation. We have made correction for the interpretation of increased RER.

6. Related to figure 3e and figure 5f, what is the reason for the increase in RER in *Ido1*-aKO and *Ahr*-aKO mice?

Responses: RER is an indicator of fuel preference. Therefore, the increase of RER in *Ido1*^{-/-}, *Ido1*-aKO and *Ahr*-aKO mice can be explained as a shift of fuel preference towards carbohydrate utilization. This

was also supported by the increased SUVmax on FDG-PET in Fig. S2m and Fig. 3f. Similar phenomena were also reported by others¹²⁻¹⁴.

12. Bapat SP, Suh JM, Fang S, et al. Depletion of fat-resident Treg cells prevents age-associated insulin resistance. *Nature*, 2015, 528(7580):137.

13. Zhang L, Avery J, Yin A, et al. Generation of functional brown adipocytes from human pluripotent stem cells via progression through a paraxial mesoderm state. *Cell stem cell*, 2020.

14. Silva VRR, Micheletti TO, Pimentel GD, et al. Hypothalamic S1P/S1PR1 axis controls energy homeostasis. *Nature Communications*, 5:4859.

7. On page 9, “in silicon” should be replaced with “in silico”.

Responses: We made correction for this typo mistake.

8. The STAT3 blot in figure 4k seems to have been modified. The authors should provide the original western blot images for all the figure panels.

Responses: We apologize for this confusion, which was likely caused by the conversion of file into word. To assure the accuracy of this result, we repeated the experiment, and the result was updated into Fig. 4j. Also, all original western blot images have been provided.

9. The words “surprisingly, interestingly, impressively, etc” are over-used in the manuscript and should be removed.

Responses: We have removed these words in the revised manuscript.

We appreciate the enthusiasm from all Reviewers. We have made necessary corrections suggested by the Reviewers. We hope that the manuscript is now ready for publication in the **Nature Communications**. Should you have additional questions, please kindly inform me.

Sincerely Yours

Cong-Yi Wang, PhD
Director and Professor, the Center for Biomedical Research
Tongji Hospital
Tongji Medical College
Huazhong University of Science and Technology
wangcy@tjh.tjmu.edu.cn

REVIEWERS' COMMENTS

Reviewer #1 (Remarks to the Author):

The authors addressed all my concerns in a very comprehensive fashion

Reviewer #2 (Remarks to the Author):

I have no other comments.

Reviewer #3 (Remarks to the Author):

The authors have sufficiently addressed the critiques raised by me and other reviewers. The manuscript has been very much improved compared to the initial submission. My only suggestion would be to include the original p-values of the statistical tests instead of the symbols.